# Serum peptidomic profiling and peptide mass fingerprinting reveal signatures associated with peroxisomal and mitochondrial pathways in MMVD-associated cardiorenal syndrome in dogs

Poonavit Pichayapaiboon[1], Sekkarin Ploypetch[2], Walasinee Sakcamduang[2], Sittiruk Roytrakul[3], Janthima Jaresitthikunchai[3], Narumon Phaonakrop[3], Wachira Trakoolchaisri[1], Ruangrat Buddhirongawatr[2*]

1 Prasu Arthorn Veterinary Teaching Hospital, Faculty of Veterinary Science, Mahidol University, Nakhon Pathom, Thailand, 2 Department of Clinical Sciences and Public Health, Faculty of Veterinary Science, Mahidol University, Nakhon Pathom, Thailand, 3 National Center for Genetic Engineering and Biotechnology, National Science and Technology, Development Agency, Pathum Thani, Thailand

* ruangrat.bud@mahidol.ac.th

## Abstract

Cardiorenal syndrome in dogs, particularly those with myxomatous mitral valve degeneration (MMVD), involves complex neurohormonal interactions, yet molecular mechanisms linking heart and kidney dysfunction remain poorly understood. This study aimed to identify serum peptide and protein signatures that differentiate disease stages and clarify pathophysiology. We hypothesized that alterations in the serum peptide mass fingerprint and protein profile would serve as sensitive biomarkers for kidney disease progression in MMVD-affected dogs. Dogs were classified into five groups following ACVIM consensus guidelines for the diagnosis and treatment of myxomatous mitral valve disease in dog and IRIS Staging of chronic kidney disease (CKD): Healthy, MMVD Stage B1, MMVD Stage C without azotemia, MMVD Stage C with azotemia, and CKD IRIS Stage 2. Serum peptide profiling was performed using Matrix-Assisted Laser Desorption/Ionization – Time-of-Flight Mass Spectrometry (MALDI-TOF MS) for peptide mass fingerprint analysis and Nanoscale Liquid Chromatography-Tandem Mass Spectrometry for protein identification. MALDI-TOF MS effectively discriminated healthy dogs from diseased groups through discrete clustering in Partial Least-Squares Discriminant Analysis plots. Proteomic profiling identified 100 statistically significant proteins, with a critical subset of shared proteins including Peroxisomal biogenesis factor 14, Glutaredoxin-2, Creatine kinase mitochondrial 2, and Selenocysteine insertion sequence-binding protein identified between the MMVD C WAZ and CKD Stage 2 groups. These proteins are associated with the arginine and proline metabolism and peroxisome pathways, which were found to be influenced by standard medications such as pimobendan, furosemide, spironolactone, benazepril, and sildenafil. These findings reflect a systemic failure of cellular redox and energetic homeostasis driven by renin angiotensin aldosterone

**Data availability statement:** All relevant data are within the paper and its Supporting information files.

**Funding:** This research project is supported by the Faculty of Veterinary Science, Mahidol University.

**Competing interests:** The authors have declared that no competing interests exist.

system activation and oxidative stress. The convergence of peroxisomal and mitochondrial dysfunction suggests a unifying mechanism for parallel cardiac and renal deterioration. These proteomic markers persist despite standard medical therapy, highlighting their potential as auxiliary tools for risk stratification and the monitoring of treatment efficacy in canine cardiorenal syndrome.

## Introduction

Cardiorenal syndrome (CRS) is a complex disease that involves the cardiovascular system, the kidneys, and numerous neurohormonal systems [1]. It leads to dysfunction in one organ that impacts another [2]. This syndrome is classified into five subtypes, including acute and chronic CRS, acute and chronic renocardiac syndrome, and secondary CRS [3]. In dogs, the prevalence of azotemia with heart disease is between 7.4 and 24.1% [4]. Myxomatous mitral valve degeneration (MMVD) is the most common acquired cardiac disease in dogs, with its incidence increasing significantly with age [5]. The affected dog suffers from a reduction of cardiac output and triggering of compensatory mechanisms including the renin angiotensin aldosterone system (RAAS) and chronic stimulation of the sympathetic nervous system. This ultimately leads to congestive heart failure (CHF) [6,7]. Numerous investigations have demonstrated that both clinical and subclinical stages of MMVD are significant risk factors for the progression of kidney disease [8–12]. The American College of Veterinary Internal Medicine (ACVIM) consensus guideline for MMVD in dogs recommends treating affected dogs with pimobendan, an angiotensin converting enzyme inhibitor (ACEi), diuretics, and sildenafil [13]. Indeed, some of these medications may aggravate pre-existing renal damage [14–16]. Therefore, monitoring for renal damage during the treatment of heart disease is crucial. Routinely, serum creatinine and serum symmetric dimethylarginine (SDMA) are used as clinical renal markers in chronic kidney disease (CKD). However, they have some limitations, including low sensitivity for renal damage and the potential to be affected by concurrent diseases [17]. Proteomic techniques have emerged as powerful tools for identifying potential clinical biomarkers for various diseases in dogs, including MMVD and CKD [18–23]. Additionally, peptide mass fingerprints (PMFs), a proteomics approach that characterizes serum peptides by comparing the pattern of mass spectra have been used in exploratory research to distinguish peptide pattern differences in canine diseases, including MMVD associated with pulmonary hypertension (PH) and oral cancers [21,24], although these applications remain investigational and are not part of current diagnostic guidelines. Despite these advancements, the application of PMFs and comprehensive peptidomic profiling in CRS remains limited. Due to the complex interaction between the heart and kidneys in CRS, peptidomic analysis may help identify biomarkers and clarify the molecular mechanisms of kidney involvement. This could support earlier diagnosis and assist in designing more effective treatment plans. This investigation aimed to identify untargeted peptidomic signatures associated with cardiorenal syndrome type 2 (CRS type 2) secondary to myxomatous mitral

valve disease (MMVD) using two complementary analytical approaches. First, matrix-assisted laser desorption ionization–time of flight mass spectrometry (MALDI-TOF MS) was used to compare peptide mass fingerprints (PMFs) among healthy dogs, dogs with non-azotemic MMVD, and dogs with azotemic MMVD representing MMVD-associated CRS type 2, with dogs with chronic kidney disease (CKD; IRIS stage 2) included primarily as a renal disease comparison group. PMF profiles were correlated with clinicopathologic data, echocardiography, and ultrasonography findings. Second, nano–liquid chromatography tandem mass spectrometry (nanoLC-MS/MS) was performed to analyze peptide amino acid sequences based on mass-to-charge ratio (m/z) and to identify overexpressed proteins associated with MMVD-related cardiorenal pathophysiology, thereby providing insight into molecular association underlying the progression of CRS type 2 in dogs with chronic mitral valve disease.

We hypothesized that specific alterations in serum PMFs and peptide profiles detectable by peptidomic analysis may serve as sensitive biomarkers for detection and risk stratification of renal dysfunction in dogs with MMVD-associated CRS type 2, independent of changes induced by standard pharmacological treatments (e.g., pimobendan, ACE inhibitors, and diuretics).

## Materials and methods

### Animals and sampling

This study was a case-control study with multiple case groups, conducted from July 2023 to July 2024. All procedures were performed in accordance with the Institutional Animal Care and Use Committee, Faculty of Veterinary Science, Mahidol University (Approval number, MUVS-2023-03-20; Approval date, 20 March 2023). No animals were euthanized for the purpose of this study, and no procedures requiring anesthesia or analgesia were performed. All samples were collected during routine clinical evaluations, and efforts were made to minimize animal discomfort. Serum samples were collected from small-breed dogs over seven years of age and weighing less than 15 kg. Each dog underwent a comprehensive health assessment, including complete physical examination, complete blood count (CBC), blood chemistry profiles, echocardiography, abdominal ultrasound, blood pressure measurement, and urinalysis. Sample size estimation was guided by feasibility and by previous exploratory proteomic studies in dogs, which typically include 8–15 animals per group [18,21,23].

The study cohorts were meticulously established to represent distinct disease and treatment states relevant to cardiorenal syndrome. The healthy group comprised dogs with no history or clinical signs of systemic diseases, demonstrating normal results across all diagnostic assessments, including CBC, blood chemistry profiles, echocardiography, abdominal ultrasound, blood pressure measurement, and urinalysis. Other groups were classified following the ACVIM consensus guidelines for MMVD and the IRIS staging system for CKD [13,14]. Dogs with subclinical MMVD (ACVIM Stage B1) were included as they represent the early stage of MMVD with no significant cardiac remodeling and, crucially, no required pharmacological treatment, allowing the identification of biomarkers of primary cardiac disease and subclinical cardiorenal crosstalk unconfounded by therapeutic intervention. Conversely, the clinical MMVD (ACVIM Stage C) group represents overt congestive heart failure and is characterized by the mandatory use of multiple pharmacological agents (pimobendan, ACEi, diuretics). Dogs classified as MMVD Stage C had a documented disease duration of at least three months and a history of clinical signs consistent with congestive heart failure, including coughing, tachypnea, dyspnea, exercise intolerance, lethargy, decreased activity, orthopnea, respiratory distress, or syncope, confirmed through medical records and clinical evaluation. Diagnosis of MMVD Stage C was further supported by echocardiographic evidence of mitral valve degeneration with mitral regurgitation and cardiac enlargement consistent with the ACVIM consensus guidelines, together with thoracic radiographic evidence of cardiomegaly and pulmonary edema indicative of congestive heart failure [13]. All MMVD Stage C dogs had previously experienced clinical signs of congestive heart failure but had been medically treated with standard therapy and remained clinically stable for at least three months prior to sample collection. This Stage C group was further divided to include the azotemic clinical MMVD group (MMVD C WAZ), which was characterized by the

presence of CKD identified through history, physical examination, blood chemistry, urinalysis, and abdominal ultrasound, exhibiting persistently elevated serum creatinine concentrations consistent with IRIS Stage 2 (1.4–2.8 mg/dL) for over three months, in the absence of systemic hypertension or proteinuria [25]. Similarly, the CKD IRIS Stage 2 group included dogs with clinical signs of CKD and serum creatinine levels within the IRIS Stage 2 range, accompanied by a substage of normotension and absence of proteinuria. The exclusion of ACVIM Stage B2 dogs was a deliberate methodological decision to create distinct, non-overlapping groups; since guidelines recommend pimobendan therapy for Stage B2, its inclusion would have introduced confounding variables related to drug effects at an intermediate stage, thus blurring the distinction necessary to definitively isolate unique peptide signatures associated with treatment-naïve disease progression versus established cardiorenal pathophysiology under standard care.

A single 5 mL blood sample was collected from each dog by venipuncture of the cephalic or saphenous vein. Serum was separated from the collected blood samples by centrifugation and immediately stored at −20 °C until further analysis. The total protein concentration in the obtained serum samples was determined using Lowry's assay, which was performed according to a standardized protocol [26].

## Complete blood count and blood chemistry profiles

CBC and blood chemistry profiles were performed on the same day as blood collection, using freshly obtained samples. Hematological parameters, including total and differential white blood cell count, red blood cell count, hemoglobin, hematocrit, mean corpuscular volume, mean corpuscular hemoglobin, mean corpuscular hemoglobin concentration, and platelet count were determined using an automated hematology analyzer (BC-5300 Auto Hematology Analyzer; Mindray, Shenzhen, China). Serum biochemical analyses included serum creatinine, blood urea nitrogen (BUN), alanine aminotransferase, alkaline phosphatase, total protein, albumin, globulin, and blood glucose, and were conducted using a blood chemistry analyzer (AU480 Chemistry Analyzer; Beckman Coulter, CA, USA).

## Radiography

Thoracic radiographs were obtained in all enrolled dogs to confirm normal thoracic anatomy in healthy controls and to exclude concurrent thoracic abnormalities (e.g., pulmonary disease, masses, anatomical defects) in all groups, including dogs with MMVD stage B1 and stage C and to evaluate for cardiomegaly and pulmonary edema. Radiographic examinations were performed using a digital radiography system (Digital diagnost C50; Philips Digital Radiography, Koninklijke Philips N.V., Thailand). The vertebral heart score (VHS) was measured on right lateral thoracic radiographs, and cardiomegaly was defined as a VHS greater than 10.5 vertebral bodies. Pulmonary edema was identified by the presence of a bronchointerstitial to alveolar lung pattern in any lung lobe of either the right or left side on thoracic radiographs [13]. Vertebral left atrial size (VLAS) was also measured for all dogs to further support radiographic assessment of left atrial enlargement in MMVD cases. Breed-specific VHS was not used due to the heterogeneous breed composition and limited numbers per breed; instead, a unified VHS cutoff was selected for consistency.

## Echocardiography

Echocardiography was performed by a single experienced sonologist using the Vivid E95 Ultra Edition system (GE Healthcare, Tokyo, Japan) with a 2.7–8 MHz phased-array transducer. All examinations were conducted while the dogs were conscious and unsedated in right and left lateral recumbency. A standard three-lead ECG was attached for cardiac cycle timing, and all measurements were obtained over three consecutive cardiac cycles and averaged. The sonologist was blinded to the group allocation of each dog to minimize observer bias. Left ventricular dimensions were indexed to body weight using allometric scaling with normalized left ventricular internal diameter in diastole (LVIDdN) calculated as: $LVIDdN = LVIDd/ BW^{0.294}$ [27], where LVIDd is left ventricular internal diameter in diastole in centimetres and BW in

kilograms. Left atrial enlargement was assessed using the left atrial to aortic ratio (LA/Ao) obtained from the right parasternal short-axis view. Chamber enlargement was defined as LA/Ao ≥ 1.6 and LVIDdN ≥ 1.7, consistent with ACVIM consensus guidelines for the diagnosis and treatment of myxomatous mitral valve disease in dogs [13]. Mitral regurgitation was evaluated based on leaflet thickening and the presence of a mosaic regurgitant jet pattern on Colour Doppler. The following echocardiographic parameters were recorded: LA/Ao, LVIDd, LVIDdN, left ventricular internal diameter in systole (LVIDs), fractional shortening (FS%), Mitral inflow velocities (E-wave) and Mitral valve peak A wave (A-wave) velocities, E/A ratio, and qualitative assessment of left atrial and left ventricular chamber size.

## Abdominal ultrasound

Abdominal ultrasonography was performed using a LOGIQ E10s system (GE Healthcare, Tokyo, Japan) with 3–10 MHz microconvex and 6–15 MHz linear probes. A single operator, blinded to group allocation, obtained transverse and longitudinal images of both kidneys to assess renal morphology and perfusion. CKD was identified based on irregular renal shape, increased cortical or medullary echogenicity, and reduced intrarenal doppler signals. Colour Doppler was used to evaluate cortical perfusion, and perfusion loss was recorded as present or absent; when present, the non-perfused area was outlined and calculated as a percentage of the total cortical area using on-screen measurement tools. Mild perfusion reduction involving approximately 20–30% of the cortex is consistent with early CKD (IRIS stage 2) [24,27,28] whereas the dogs with more extensive loss (>50–60%) were excluded from the study.

## Blood pressure measurement

Indirect systolic blood pressure was measured using a Doppler device (Model 811-BL, Parks Medical Electronics, Aloha, Oregon, USA). Measurements were obtained with dogs in sternal or lateral recumbency after a 5–10 minutes acclimatization period in a quiet room at a stable ambient temperature. Blood pressure was recorded from either the tail base or a distal limb using a cuff width equal to 30–40% of the circumference. The first measurement was discarded, and additional readings were repeated until five consistent values (variation ≤20 mmHg) were obtained; readings associated with movement, panting, or tachycardia were excluded. The mean of the five accepted systolic blood pressure measurements was used. Following IRIS guidelines, dogs with a mean systolic blood pressure >140 mmHg were classified as hypertensive [25] and excluded from the study.

## Urinalysis

Urine samples were collected by free-catch voiding during spontaneous urination to minimize stress and avoid invasive procedures. Physical and chemical urinalysis was performed using an automated urine analyzer (Mission U500, ACON Laboratories, San Diego, CA, USA), and microscopic sediment examination followed previously described methods [29]. The urine protein to creatinine ratio (UPCR) was measured using a chemistry analyzer (AU480, Beckman Coulter, Brea, CA, USA). In accordance with IRIS guidelines, dogs were excluded if they had proteinuria (UPCR >0.5) or borderline proteinuria (UPCR 0.2–0.5), as either condition may indicate renal injury or tubular dysfunction that could confound proteomic comparisons.

## Statistical analysis

Statistical analyses were performed using SPSS (version 28, IBM, Armonk, NY, USA). The numerical data were assessed for normality using the Shapiro–Wilk test. If the data passed the normality test, one-way ANOVA was conducted. For data that did not meet the normality assumption, the Kruskal–Wallis test was employed. The Bonferroni test was used for post hoc analysis. Categorical data were analyzed by Chi-square test. A p-value less than 0.05 was considered statistically significant.

## Proteomic analysis for cluster analysis and searching candidate peptides

**Matrix-Assisted Laser Desorption/Ionization Time-of-Flight Mass Spectrometry (MALDI-TOF MS).** The serum peptides were subjected to fractionation using ultrafiltration membranes with molecular weights of 10 kDa (Amicon Ultra15, Merck Millipore Ltd., Darmstadt, Germany). We used the < 10 kDa peptide fractions obtained by ultrafiltration of the serum samples. The peptides were collected and kept at −20 °C until use. Peptides (0.1 μg/μL) were prepared in 0.1% trifluoroacetic acid (TFA) and subsequently mixed with a MALDI matrix solution composed of 10 mg/mL α-cyano-4-hydroxycinnamic acid (CHCA) in 100% acetonitrile (ACNI) with 5% TFA (1:1, v/v). The resulting mixture was directly spotted onto a MALDI steel target plate (MTP 384 ground steel, JEOL, Japan). To minimize sample preparation bias, eight replicates were analyzed. Following air drying, mass spectrometry was performed using a JMS-S3000 SpiralTOF (JEOL, Japan) in linear positive mode, covering a mass range of 1,000–6,000 Da. Each sample was subjected to 500 accumulated laser shots at 50 Hz. Mass spectra were processed using msTornado Analysis version 1.15, with peptide profiles and partial least squares-discriminant analysis (PLS-DA) employed for data interpretation.

To assess whether PMF and peptide profiles differed among groups healthy dogs, MMVD stage B1 dogs, dogs with MMVD stage C without azotemia, dogs with MMVD stage C with azotemia, and dogs with CKD IRIS stage 2 hierarchical clustering analysis was performed using the Sartorius Stedim Data Analytics' prediction engine (SIMCA®). PLS-DA was conducted using spectral data from eight replicates per dog in each group. To further confirm peptide profile similarities within each group, serum samples were pooled and subjected to mass spectrometry (MS) analysis. Pooled serum peptides (0.1 μg/μL) were prepared in 0.1% trifluoroacetic acid (TFA), mixed with a MALDI matrix solution, and spotted onto an MTP384 target plate (Bruker Daltonics) for air drying. The pooled sample from each group was spotted in technical triplicates for a total of 32 spots analyzed across all groups, with the data used to assess reproducibility and prepare for PLS-DA and outlier detection. MS spectra were analysed as previously described [30].

**Nanoscale liquid chromatography coupled with tandem mass spectrometry (nanoLC-MS/MS).** Peptidomics analysis was performed as previously described by Ploypetch et al. (2025) using an Ultimate 3000 Nano/Capillary LC System (Thermo Scientific, UK) coupled to a ZenoTOF 7600 mass spectrometer (SCIEX, Framingham, MA, USA) [30]. In each group, 5 μg of peptides from each sample were pooled and reduced with a disulfide-containing compound and cysteine residues. Subsequently, the samples were incubated in the dark at 25 °C for 45 minutes. Samples were dissolved in 0.1% formic acid (FA) and subjected to nanoLC-MS/MS. To ensure the reliability and confidence of our quantification, each pooled group was analyzed in technical triplicate.

One μL of peptide was first enriched on a C18 μ-Precolumn (300 μm i.d. × 5 mm, C18 PepMap 100, 5 μm, 100 Å) and then separated on a 75 μm inner diameter. × 15 cm analytical column (Acclaim PepMap RSLC C18, 2 μm, 100 Å, nanoViper, Thermo Scientific, UK) at 60 °C, using a 30-minutes gradient of 5–55% acetonitrile in 0.1% FA at 0.30 μL/min. The mass spectrometer operated in positive ion mode with data-dependent acquisition (Top 50), using a dynamic exclusion of 12 seconds and MS2 scan range of 100–1,800 *m/z*. Key settings included source gas 1 at 8 psi, curtain gas at 35 psi, source temperature 200 °C, spray voltage 3,300 V, and a Zeno trap threshold of 150,000 cps. The MS cycle time was set to 3.0 seconds.

## Bioinformatics and statistical analysis

Peptide and protein quantifications were performed using MaxQuant (version 2.2.0.0), with the Andromeda search engine matching MS/MS spectra to the *Canis familiaris* database from UniProt (download on December 9, 2024) Label-free quantification followed standard MaxQuant parameters, allowing up to two missed cleavages, a 0.6 Da mass tolerance for the primary search, carbamidomethylation of cysteine as a fixed modification, and methionine oxidation and N-terminal acetylation as variable modifications. For protein identification, a stringent filtering process was applied. Only peptides with a minimum length of seven amino acids were considered, and a minimum of two unique peptides per protein was required for identification. A 1% false discovery rate (FDR) at the protein level was maintained using a decoy (reversed sequence)

database search. To further ensure identification accuracy, a maximum of five modifications per peptide was permitted. The *Canis familiaris* proteome FASTA file was retrieved from UniProt on December 9, 2024.

Multivariate analysis, including ANOVA, PLS-DA, and heatmap, was conducted in MetaboAnalyst 6.0 (http://www.metaboanalyst.ca) [31]. Peptides were considered significantly altered if they met both the following conditions: FDR P-value < 0.05 and Fisher's least significant difference < 0.05. Protein list comparison among different sample groups was displayed using Jvenn diagram (https://jvenn.toulouse.inrae.fr/app/example.html) [32]. Candidate peptides were further analyzed for protein–protein and protein–drug interactions using STITC (http://stitch.embl.de/) [33]. Gene ontology enrichment analysis was performed using ShinyGo software (version 0.77) (https://bioinformatics.sdstate.edu/go77/) to classify the biological processes, cellular components, and molecular functions of the identified proteins [34]. Pathway analysis was performed using the Kyoto Encyclopedia of Genes and Genomes (KEGG) to investigate interactions with pimobendan, furosemide, spironolactone, ACEi, and sildenafil

## Results

### Sample description data

The demographic characteristics of the 64 dogs enrolled in this study are presented in Table 1. The study population consisted of five groups: 15 clinically healthy individuals (Healthy), 10 dogs with MMVD at stage B1 (MMVD B1), 15 dogs with MMVD stage C without azotemia (MMVD C WOAZ), 13 dogs with MMVD stage C with azotemia (MMVD C WAZ), and 11 dogs with CKD at IRIS stage 2 (CKD stage 2). Age comparisons revealed that healthy dogs were significantly younger than dogs in all disease groups with a statistically significant difference (p < 0.01). In contrast, no significant differences were detected in body weight among the five groups (p = 0.98). The sex distribution within each group was approximately balanced between male and female dogs. No medication was administered to dogs in the Healthy, MMVD B1, and CKD stage 2 groups prior to sample collection. Dogs with MMVD stage C received standard cardiovascular therapy, which included one or more of the following: pimobendan, furosemide, spironolactone, ACEi, and sildenafil (S1 Table).

Hematologic and blood chemistry values for the Healthy, MMVD B1, and MMVD C WOAZ groups (S2 Table) were within normal reference ranges. Dogs in the MMVD C WAZ and CKD stage 2 groups showed mild azotemia, consistent

**Table 1. The demographic characteristics of the 64 dogs enrolled in this study.**

| Group | Number of samples | Ages (years) mean±SD | Body weight (kg) mean±SD | Sex (no. of samples) | Breed (no. of samples) |
|---|---|---|---|---|---|
| Healthy | 15 | 8.13±0.9 | 6.20±2.14 | M (4), Mc (4), F (4), and Fs (3) | Chihuahua (3), Pomeranian (3), Shih tzu (3), Mixed (3), Boston terrier (1), Jack Russell terrier (1), and Dachshund (1) |
| MMVD B1 | 10 | 11.3±1.8** | 6.02±1.28 | M (3), Mc (1), F (4), and Fs (2) | Chihuahua (4), Pomeranian (4), Jack Russell terrier (1), and Pekingenes (1) |
| MMVD C WOAZ | 15 | 11.8±2.8** | 5.82±1.48 | M (4), Mc (3), F (6), and Fs (2) | Chihuahua (5), Pomeranian (3), Toy poodle (4), Shih tzu (1), and Mixed (2) |
| MMVD C WAZ | 13 | 12.9±2.0** | 6.04±1.37 | M (4), Mc (3), F (3), and Fs (3) | Chihuahua (3), Pomeranian (1), Toy poodle (3), Shih tzu (4), and Mixed (2) |
| CKD stage 2 | 11 | 11.54±3.7** | 6.18±2.29 | M (1), Mc (4), F (2), and Fs (4) | Chihuahua (1), Toy poodle (1), Shih tzu (3), Mixed (3), Beagle (1), Yorkshire terrier (1), and Pug (1) |

Abbreviations: F, female; Fs, spayed female; M, male; Mc, neutered male; Healthy, healthy control; MMVD B1, myxomatous mitral valve disease stage B1; MMVD C WOAZ, MMVD stage C without azotemia; MMVD C WAZ, MMVD stage C with azotemia; CKD stage 2, chronic kidney disease at IRIS stage 2. Statistical differences among groups were assessed using one-way ANOVA followed by Bonferroni post hoc testing. Superscript letters indicate statistically significant differences between groups (P < 0.01).

with IRIS stage 2 classification. Because anemia, hypoalbuminemia, and uremia can influence serum peptide profiles through effects on protein metabolism, oxidative stress, and inflammatory activity, these variables were evaluated. No enrolled dogs demonstrated clinically significant anemia, marked hypoalbuminemia, or severe uremia based on complete blood count and serum biochemistry results. Thoracic radiographs revealed normal findings in the Healthy, MVD B1, and CKD stage 2 groups. However, cardiomegaly, defined as a VHS > 10.5, was observed in both MMVD C WOAZ and MMVD C WAZ groups, with or without concurrent pulmonary edema (S3 Table).

Thoracic radiographic and echocardiographic indices for the five study groups Healthy, MMVD B1, MMVD C WOAZ, MMVD C WAZ, and CKD stage 2 are summarized in S3 Table. Radiographically, both VHS and VLAS increased significantly in the MMVD groups, particularly in MMVD C WOAZ and MMVD C WAZ, reflecting cardiomegaly and left atrial enlargement (both P < 0.001). Healthy and CKD stage 2 dogs exhibited VHS and VLAS values within normal ranges, consistent with the absence of primary cardiac disease.

Echocardiographic assessment revealed pronounced chamber enlargement in the clinical MMVD groups. LVIDd, LVIDs, normalized LVIDdN, and LA dimensions were significantly increased in MMVD C WOAZ and MMVD C WAZ compared with Healthy, MMVD B1, and CKD stage 2 groups (all P < 0.001). Systolic function, assessed by FS%, did not differ significantly across groups. E-wave and A-wave were elevated in the clinical MMVD groups (P < 0.05), while MR Vmax and TR Vmax were present only in MMVD C dogs, confirming hemodynamically significant regurgitation. CKD stage 2 dogs showed no evidence of structural cardiac enlargement and clustered closely with healthy controls in most echocardiographic parameters. Dogs in the MMVD stage B1 group demonstrated mitral regurgitation without evidence of cardiomegaly. Both the MMVD C WOAZ and MMVD C WAZ groups exhibited left atrial and left ventricular enlargement, consistent with a history of pulmonary edema in some individuals (Fig 1).

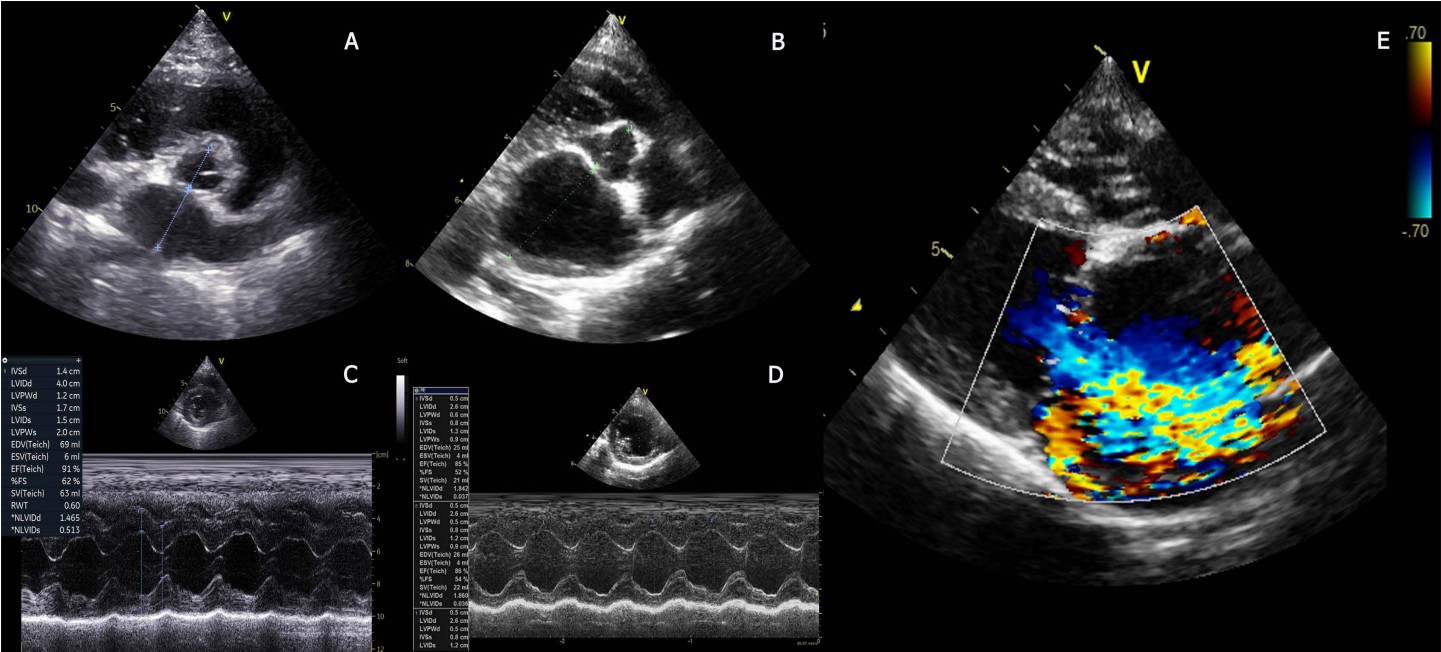

**Fig 1. Echocardiographs of the enrolled dogs.** Right parasternal shot-axis view of the left atrium level of MMVD stage B1 dogs **(A)** and MMVD stage C dogs **(B)**, M-mode study of left ventricle at the papillary muscle of MMVD stage B1 **(C)** and MMVD stage C dogs **(D)**.Right parasternal long axis view with a Colour Doppler study of the left atrium of MMVD stage C dogs **(E)**.

Abdominal ultrasonography of MMVD C WAZ and CKD stage 2 groups demonstrated mildly decreased renal perfusion, irregular renal contours, and increased echogenicity of both the cortical and medullary regions, as shown in Fig 2. These sonographic abnormalities were not observed in healthy controls, MMVD B1, MMVD C WOAZ groups.

Indirect systolic blood pressure did not differ significantly among groups (P=0.599) and remained within the normotensive range in all dogs. Median systolic blood pressure values ranged from 129 to 131 mmHg across the five groups (S3 Table)

Urinalysis parameters across the five study groups Healthy, MMVD B1, MMVD C WOAZ, MMVD C WAZ, and CKD stage 2 are summarized in S4 Table. Urine colour and clarity showed no significant differences among groups (P=0.28 and P=0.89, respectively). Most samples were yellow to pale yellow with clear to slightly cloudy appearance. Chemical dipstick findings, including protein, glucose, ketones, bilirubin, hemoglobin, urobilinogen, nitrite, and leukocytes, were largely unremarkable across all groups, with no statistically significant differences detected (P>0.05). A notable difference was observed in urine concentrating ability.

Urine specific gravity (USG) decreased significantly in all disease groups compared with healthy dogs (P<0.001). Healthy dogs demonstrated a median USG of 1.041, while MMVD B1, MMVD C WOAZ, MMVD C WAZ, and CKD stage 2 groups showed progressive reductions in USG, with MMVD C WAZ exhibiting the lowest values. Urine pH did not differ significantly (P=0.062), and crystal type distribution was comparable among groups. Sediment examination (RBCs and WBCs per high-power field) also showed no significant differences. Urine creatinine and urine protein concentrations did not vary significantly (P=0.961 and P=0.862, respectively). In contrast, UPCR differed significantly among groups (P<0.001). Healthy and MMVD B1 dogs had UPCR values within the expected non-proteinuric range (median 0.09–0.10). Dogs in MMVD C WOAZ, MMVD C WAZ, and CKD stage 2 groups exhibited modest but significantly higher UPCR values (median 0.15), though values remained below the threshold for clinically relevant proteinuria (S4 Table).

## Analysis of serum peptides by MALDI-TOF MS

In the 2D PLS-DA plot, the spectral data from the four replicates of each sample exhibited a distinct clustering pattern among the five groups using SIMCA®. Notably, the control group was clearly separated from the others, while the MMVD B1, MMVD C WAZ, MMVD C WOAZ, and CKD stage 2 groups clustered closely together, indicating a higher degree of similarity in their peptide profiles (Fig 3A). To ensure consistency within groups and validate the related peptide profiles, 32 replicates of pooled samples were analyzed, confirming homogeneity within each group (Fig 3B). The distinct PMFs ranged from 1 to 6 kDa of pooled samples among groups (Fig 3C).

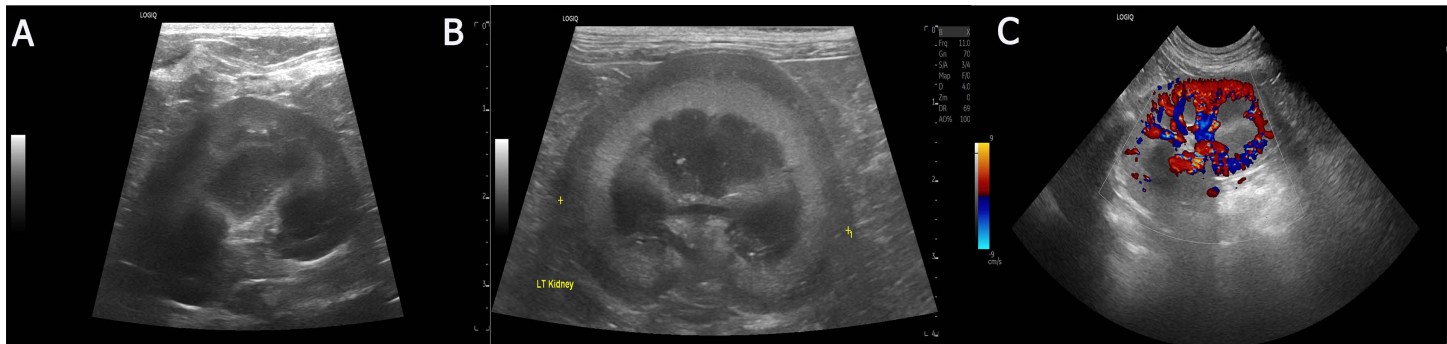

**Fig 2. Ultrasonographic results of the enrolled dogs.** Sagittal view of the kidney of dog in the healthy (A) and CKD IRIS stage 2 group **(B)**. Colour Doppler image of the kidney showing intrarenal vascular perfusion. Red and blue signals represent blood flow direction relative to the transducer. Patchy and reduced Doppler signal intensity indicates diminished cortical perfusion, a pattern consistent with early CKD-related vascular changes **(C)**.

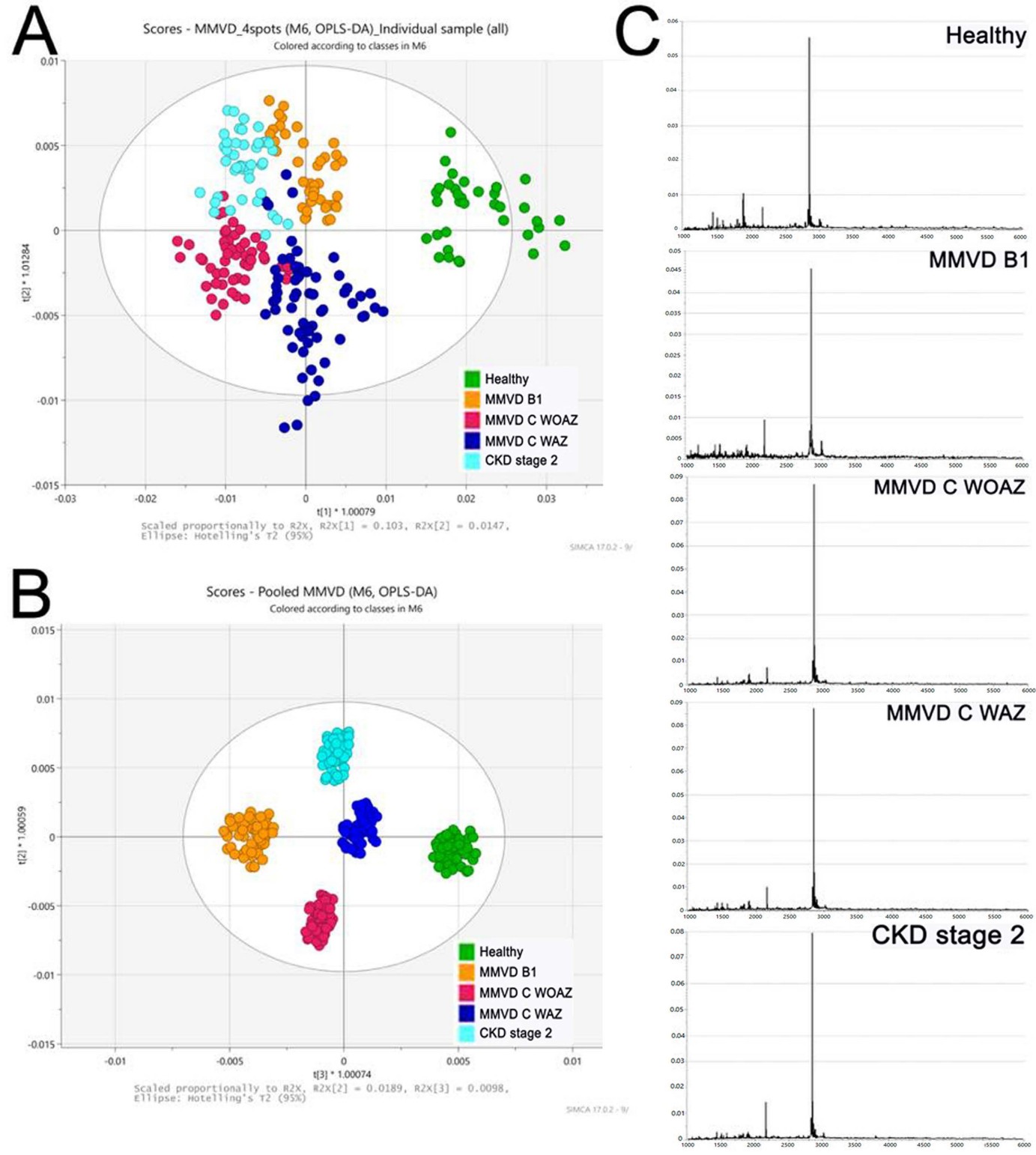

**Fig 3. Partial least squares discriminant analysis (PLS-DA) and serum peptide mass fingerprints of dogs with myxomatous mitral valve disease and chronic kidney disease.** Two-dimensional scatterplot of the results of PLS-DA of individual **(A)** and pooled **(B)** serum samples from the Healthy, myxomatous mitral valve disease stage B1 (MMVD B1), MMVD stage C without azotemia (MMVD C WOAZ), MMVD stage C with azotemia

(MMVD C WAZ), and CKD IRIS stage 2 groups (CKD stage 2), analyzed by matrix-assisted laser desorption/ionization time-of-flight mass spectrometry (MALDI-TOF MS). **(C)** Peptide mass fingerprints (PMFs) of serum peptides in the mass range of 1,000–6,000 Da.

### Peptide and protein identification by nanoLC-MS/MS

A total of 19,405 proteins from all pooled samples were analyzed using in-solution digestion followed by nanoLC-MS/MS (details in S5 Table). The distribution and overlap of individual proteins across the five experimental groups Healthy, MMVD B1, MMVD C WOAZ, MMVD C WAZ, and CKD Stage 2 were visualized using a Venn diagram (Fig 4A). A group of 510 common proteins was identified, suggesting they may represent a specific subset with shared characteristics or biological functions relevant to both the MMVD C WAZ and CKD stage 2 conditions (S6 Table). One hundred proteins were found to be statistically significant based on an ANOVA $p < 0.05$ (S7 Table). This significance suggests these proteins play a crucial role in the disease's pathophysiology. Crucially, six common proteins were identified as being present in both MMVD C WAZ and CKD stage 2 groups and were among the 100 statistically significant proteins. These six proteins are peroxisomal membrane protein PEX14 (PEX14), glutaredoxin-2 (GLRX2), creatine kinase S-type (CKMT2), SECIS binding protein 2 (SECISBP2), LRRCT domain-containing protein, and histone H4 transcription factor (HINFP). These findings are further detailed in Fig 4B. and Table 2.

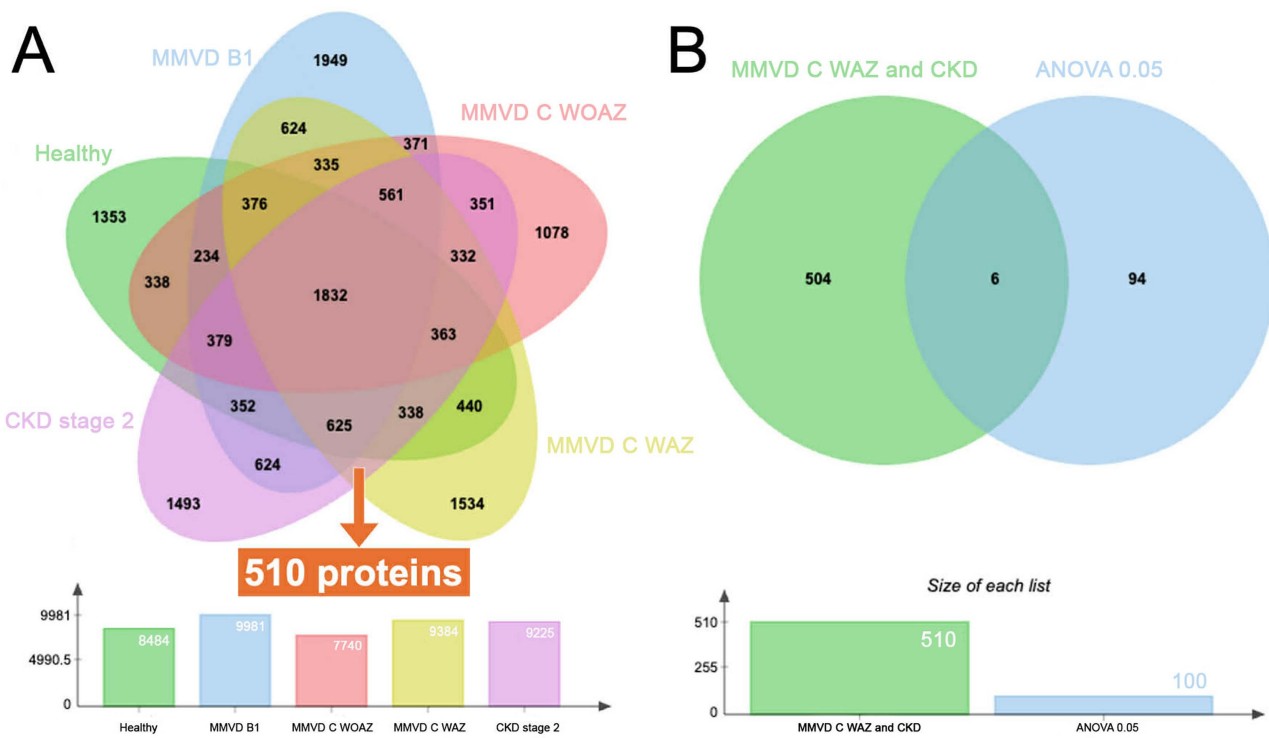

**Fig 4. Comparative serum proteomic profiles across Healthy, myxomatous mitral valve disease (MMVD), and chronic kidney disease IRIS stage 2 groups (CKD stage 2). (A)** Five-way Venn diagram and bar chart of total protein identification counts. The Venn diagram illustrates the number of unique and overlapping proteins identified in five different groups: Healthy, MMVD B1, MMVD C WOAZ, MMVD C WAZ, and CKD stage 2. Numbers within overlapping regions indicate common proteins shared between those specific groups. The number 510 highlights the proteins found exclusively in common between the CKD stage 2 and MMVD C WAZ groups. **(B)** Two-way Venn diagram and bar chart of significant proteins shared between MMVD C WAZ and CKD stage 2 groups. The Venn diagram compares this list of 510 proteins (labeled "MMVD C WAZ and CKD") with a list of proteins filtered using ANOVA at 0.05. The central overlap of 6 indicates the number of proteins that are both common to MMVD C WAZ and CKD stage 2 that meet the specific statistical criterion for significance.

**Table 2. Significant differences in peptide alterations between the MMVD C WAZ and CKD stage 2 groups based on biological process involvement, and molecular function, using nanoLC-MS/MS data with FDR, and -log10 P-value<.05.**

| Protein IDs | Protein name | Peptide sequence (Peptide count) | Gene name | - log10 P-value | FDR | Biological process | Molecular function |
|---|---|---|---|---|---|---|---|
| A0A8I3RR95 | Peroxisomal membrane protein PEX14 (Peroxin-14) | DEEDEEDEDVSHVD;PEG-STAAY;PEGSTAAYHLLG-PQEEGQG (3) | PEX14 | 3.9053 | 0.0191 | cellular response to reactive oxygen species, microtubule anchoring, peroxisome transport along microtubule, protein import into peroxisome matrix, docking, protein import into peroxisome matrix, protein import into peroxisome matrix translocation, | beta-tubulin binding identical protein binding, microtubule binding, protein transmembrane transporter activity, signaling receptor binding, transcription corepressor activity |
| A0A8P0NCH8 | Glutaredoxin-2, mitochondrial | AAGAAASG;AALRATRP;LF-HDMNVKY (3) | GLRX2 | 3.6283 | 0.0297 | apoptotic process, cell differentiation, cell redox homeostasis, and glutathione metabolic process | 2 iron, 2 sulfur cluster binding, metal ion binding |
| A0A8C0NTJ2 | creatine kinase S-type | NGYDPRVMKH;PRLSKD-PRFPKI (2) | CKMT2 | 3.6983 | 0.0267 | phosphocreatine biosynthetic process | ATP binding, creatine kinase activity |
| A0A8I3PA82 | SECIS binding protein 2 | AKKPTSLKKIILKER-QERKQQRLQ;DTKALFKKK-T;LREVLKHLKLRKLK;PVKA-KTKRRLVLGLREVLKHLK-LR;SSELALHPCS;TAEGE-GVVRSTDAVE;VGIFSY-DGAQVSTERLQSPTRP-CW;VLGLREVLKHLKLRKLK;-VLKHLKLRKLK (9) | SECISBP2 | 3.4596 | 0.0408 | forebrain neuron development, mRNA stabilization, RNA catabolic process, and selenocysteine incorporation | selenocysteine insertion sequence binding |
| A0A8C0Z0W0 | LRRCT domain-containing protein | DPGEAAAG;FFGLVLALIGLI-F;GEAAAGEAAAAG;GG-DADPGE;RRLRCASP (5) | LRRCT | 4.8661 | 0.0032 | chemical synaptic transmission, establishment of protein localization, and synaptic membrane adhesion | protein phosphatase inhibitor activity |
| A0A8I3MHT9 | Histone H4 transcription factor | EEDDDEDPLEEEFSCL-WQEC;EWFYRHVE;IKSHY-RKVHEG;RYKEHEDGYM-RLQ;VVLCGWKGCTCTFK (5) | HINFP | 3.3947 | 0.0433 | DNA damage checkpoint signaling, DNA repair, DNA-templated transcription, establishment of protein localization, G1/S transition of mitotic cell cycle, in utero embryonic development, myoblast differentiation, negative regulation of gene expression, positive regulation of gene expression | chromatin binding, DNA-binding transcription activator activity, RNA polymerase II-specific, DNA-binding transcription repressor activity, RNA polymerase II-specific, enzyme binding, histone binding, metal ion binding, RNA polymerase II cis-regulatory region sequence-specific DNA binding |

Abbreviations: FDR, false discovery rate; N/A, not applicable.

## Molecular mechanism and interaction analysis

To better understand the molecular mechanisms associated with the proteins that changed across the five groups, particularly the proteins common to the MMVD C WAZ and CKD stage 2 conditions, researchers conducted several integrated analyses: Protein-protein and protein-drug interaction network analysis and pathway enrichment analysis. This focus was specifically placed on the six upregulated proteins that were identified as both statistically significant (by ANOVA) and

common to MMVD C WAZ and CKD stage 2 (by Jvenn diagram). The functions of candidate peptides, including their involvement in disease pathophysiology, pathogenesis, and mechanisms of action were explored to identify prognostic biomarkers and potency assays for dogs that received treatment for cardiac diseases. The analysis also investigated the association between these six proteins and common cardiogenic drugs used for treatment, including pimobendan, furosemide, spironolactone, ACEi, and sildenafil. The protein-drug interaction network results (Fig 5) revealed distinct interaction patterns (accessed on December 5, 2025) using Stitch EMBL. Several upregulated proteins demonstrated multiple connections within the network, indicating interactions with all five tested drugs. This interaction for GLRX2 and CKMT2, which exhibited a strong relationship level in the protein interaction network (with an edge confidence >0.7), was present for SECISBP2 (with an edge confidence <0.4) with all drugs and functional partners, such as creatinine, creatinine phosphate, MgADP, MgATP, and reduced glutathione (Fig 5). The upregulated protein PEX14 was associated only with pimobendan (Fig 5A), while HINFP showed no interaction with any of the tested drugs (Fig 5A–5E).

### Pathway enrichment analysis: Cardio-renal syndrome links

The KEGG pathway enrichment analysis using ShinyGO 0.77 (accessed on December 7, 2025), which was performed on the six significant proteins whose interactions with cardiogenic drugs are visualized in the protein-protein interaction networks (Fig 5A–5E), strongly suggests that the observed protein alterations and drug interactions are linked to the pathophysiology of CRS (Table 3). Crucially, the analysis highlights several pathways that appeared consistently across multiple drug interactions, underscoring their systemic role in both cardiac and kidney dysfunction. Peroxisome and arginine and proline metabolism pathways were associated with all five drugs (pimobendan, furosemide, spironolactone, benazepril, and sildenafil), pointing toward fundamental metabolic components, such as altered nitrogen metabolism, relevant to both CKD and cardiovascular health. Furthermore, the analysis reveals pathways highly characteristic of CRS, including the RAAS and renin secretion, which were linked to interactions involving furosemide (Fig 5B), spironolactone (Fig 5C), and benazepril (Fig 5D). The presence of diabetic cardiomyopathy (associated with Fig 5B–5D) and the aldosterone regulated sodium reabsorption pathway (Fig 5C) emphasizes the critical cardiovascular and renal regulatory components, respectively. Finally, the cGMP-PKG signaling pathway (Fig 5E) highlights an impact on vascular tone. In summation, the consistent enrichment of pathways governing the RAAS, renal regulation, cellular metabolism, and cardiovascular health provide robust evidence that the significant proteins and their drug interactions are intricately linked to the multiorgan dysfunction characteristic of cardiorenal syndrome. The biological processes and molecular functions of the protein alterations and all five drug interactions were demonstrated in S8 and S9 Tables.

## Discussion

This study investigated the serum peptide profiles in healthy dogs, as well as in those diagnosed with MMVD stage B1 (MMVD B1), MMVD stage C (with and without azotemia) (MMVD C WOAZ and MMVD C WAZ), and CKD classified as IRIS stage 2 (CKD stage 2). We did not include MMVD stage B2 dogs due to the heterogeneous nature of this transitional stage, variability in cardiomegaly severity, and the frequent initiation of pimobendan therapy at this stage. Including B2 dogs would likely increase within group variability and reduce discriminatory power in peptide-based clustering. Future studies specifically targeting the B1 to B2 transition would be valuable for evaluating early peptide changes preceding overt cardiac remodelling.

The medication distribution across groups reflects expected clinical management for each disease stage and provides important context for interpreting the proteomic findings. As anticipated, dogs with MMVD C both azotemic and non-azotemic were uniformly treated with pimobendan and diuretics, consistent with ACVIM consensus guidelines for the diagnosis and treatment of myxomatous mitral valve disease in dogs [13]. The concurrent use of ACEi and spironolactone in many MMVD C dogs reflects attempts to mitigate RAAS activation, reduce afterload, and manage volume overload [35]. Importantly, Healthy, MMVD B1, and CKD stage 2 dogs were medication-naïve with respect to cardiovascular drugs. This distinction helps ensure that observed peptide alterations in these groups are attributable to underlying

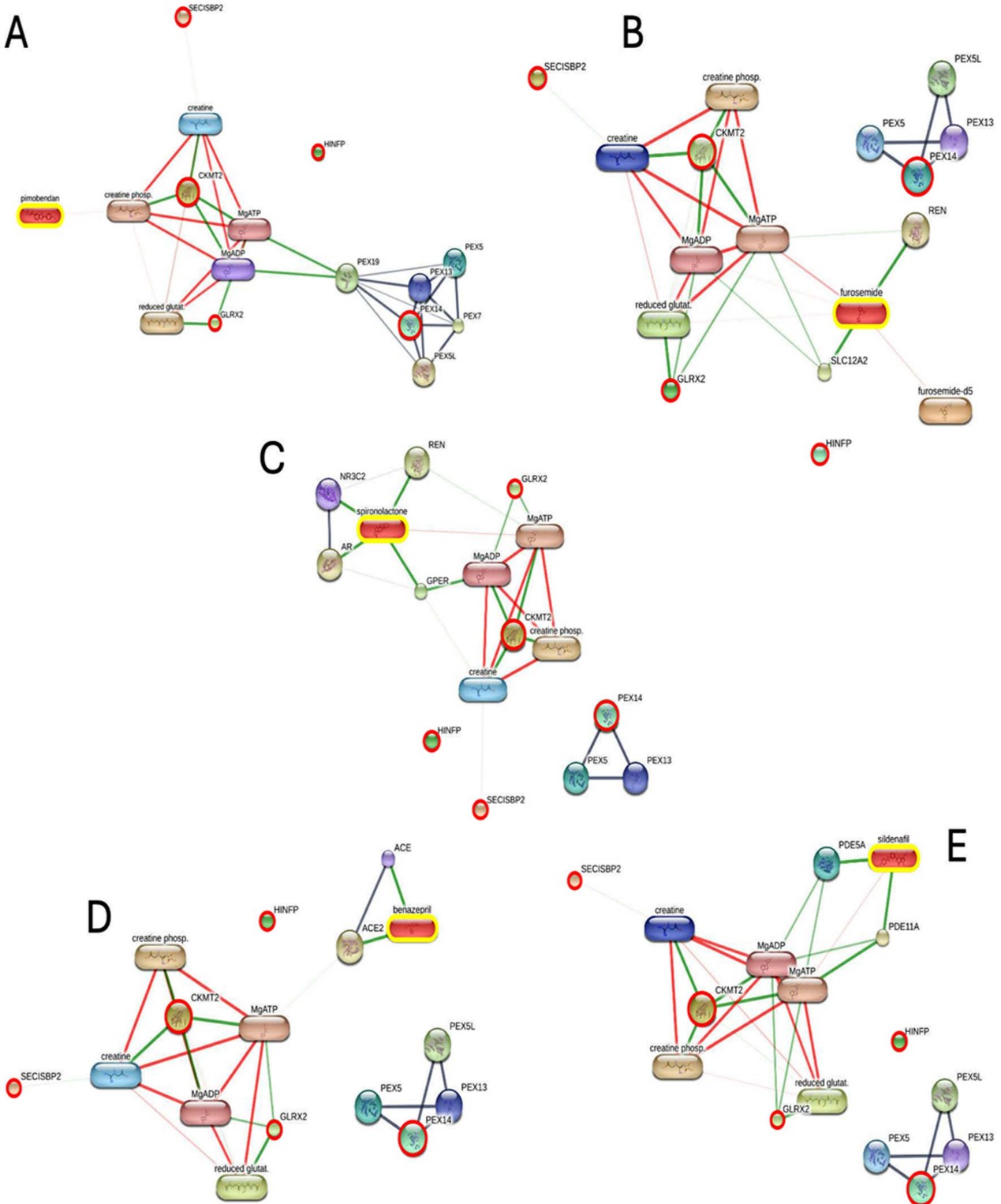

**Fig 5. Protein–drug interaction network associated with oxidative stress and peroxisomal regulation in canine cardiorenal syndrome.** Red circles: peroxisomal membrane protein PEX14 (PEX14), glutaredoxin-2 (GLRX2), creatine kinase S-type (CKMT2), SECIS binding protein 2 (SECISBP2), LRRCT domain-containing protein, and histone H4 transcription factor (HINFP). Yellow boxes: pimobendan, spironolactone, furosemide, benazepril, sildenafil.

**Table 3. Protein-drug pathway enrichment analysis performed using the Kyoto Encyclopedia of Genes and Genomes (KEGG).**

| Drug | Enrichment FDR | Genes number | Pathway Genes | Fold Enrichment | Pathway | URL | Genes name |
|---|---|---|---|---|---|---|---|
| Pimoben-dan | 0.0000 | 5 | 76 | 147.6535 | Peroxisome | http://www.genome.jp/kegg-bin/show_pathway?cfa04146 | PEX7, PEX5L, PEX19, PEX5, PEX14 |
| | 0.0497 | 1 | 45 | 49.8741 | Arginine and proline metabolism | http://www.genome.jp/kegg-bin/show_pathway?cfa00330 | CKMT2 |
| | 0.0628 | 1 | 86 | 26.0969 | Small cell lung cancer | http://www.genome.jp/kegg-bin/show_pathway?cfa05222 | SECISBP2 |
| Furosemide | 0.0000 | 3 | 76 | 88.59211 | Peroxisome | http://www.genome.jp/kegg-bin/show_pathway?cfa04146 | PEX5L, PEX5, PEX14 |
| | 0.0510 | 1 | 23 | 97.57971 | Renin-angiotensin system | http://www.genome.jp/kegg-bin/show_pathway?cfa04614 | REN |
| | 0.0538 | 1 | 45 | 49.87407 | Arginine and proline metabolism | http://www.genome.jp/kegg-bin/show_pathway?cfa00330 | CKMT2 |
| | 0.0538 | 1 | 64 | 35.06771 | Renin secretion | http://www.genome.jp/kegg-bin/show_pathway?cfa04924 | REN |
| | 0.0538 | 1 | 71 | 31.61033 | Salivary secretion | http://www.genome.jp/kegg-bin/show_pathway?cfa04970 | SLC12A2 |
| | 0.0538 | 1 | 85 | 26.40392 | Pancreatic secretion | http://www.genome.jp/kegg-bin/show_pathway?cfa04972 | SLC12A2 |
| | 0.0538 | 1 | 86 | 26.0969 | Small cell lung cancer | http://www.genome.jp/kegg-bin/show_pathway?cfa05222 | SECISBP2 |
| | 0.1019 | 1 | 190 | 11.81228 | Diabetic cardiomyopathy | http://www.genome.jp/kegg-bin/show_pathway?cfa05415 | REN |
| Spironolac-tone | 0.0059 | 2 | 76 | 59.0614 | Peroxisome | http://www.genome.jp/kegg-bin/show_pathway?cfa04146 | PEX5, PEX14 |
| | 0.0477 | 1 | 45 | 49.87407 | Arginine and proline metabolism | http://www.genome.jp/kegg-bin/show_pathway?cfa00330 | CKMT2 |
| | 0.0477 | 1 | 23 | 97.57971 | Renin-angiotensin system | http://www.genome.jp/kegg-bin/show_pathway?cfa04614 | REN |
| | 0.0477 | 1 | 35 | 64.12381 | Aldosterone-regulated sodium reabsorption | http://www.genome.jp/kegg-bin/show_pathway?cfa04960 | NR3C2 |
| | 0.0477 | 2 | 483 | 9.293306 | Pathways in cancer | http://www.genome.jp/kegg-bin/show_pathway?cfa05200 | SECISBP, AR |
| | 0.0563 | 1 | 64 | 35.06771 | Renin secretion | http://www.genome.jp/kegg-bin/show_pathway?cfa04924 | REN |
| | 0.0597 | 1 | 91 | 24.663 | Prostate cancer | http://www.genome.jp/kegg-bin/show_pathway?cfa05215 | AR |
| | 0.0597 | 1 | 86 | 26.0969 | Small cell lung cancer | http://www.genome.jp/kegg-bin/show_pathway?cfa05222 | SECISBP2 |
| | 0.0657 | 1 | 113 | 19.86136 | Oocyte meiosis | http://www.genome.jp/kegg-bin/show_pathway?cfa04114 | AR |
| | 0.0864 | 1 | 167 | 13.43912 | Chemical carcinogenesis | http://www.genome.jp/kegg-bin/show_pathway?cfa05207 | AR |
| | 0.0890 | 1 | 190 | 11.81228 | Diabetic cardiomyopathy | http://www.genome.jp/kegg-bin/show_pathway?cfa05415 | REN |
| Benazepril | 0.0001 | 3 | 76 | 88.59211 | Peroxisome | http://www.genome.jp/kegg-bin/show_pathway?cfa04146 | PEX5L, PEX5, PEX14 |
| | 0.0003 | 2 | 23 | 195.1594 | Renin-angiotensin system | http://www.genome.jp/kegg-bin/show_pathway?cfa04614 | ACE2, ACE |
| | 0.0178 | 2 | 231 | 19.43146 | Coronavirus disease | http://www.genome.jp/kegg-bin/show_pathway?cfa05171 | ACE2, ACE |

*(Continued)*

**Table 3.** (Continued)

| Drug | Enrichment FDR | Genes number | Pathway Genes | Fold Enrichment | Pathway | URL | Genes name |
|------|---------------|--------------|---------------|-----------------|---------|-----|------------|
| | 0.0571 | 1 | 45 | 49.87407 | BenazeprilArginine and proline metabolism | http://www.genome.jp/kegg-bin/show_pathway?cfa00330 | CKMT2 |
| | 0.0571 | 1 | 64 | 35.06771 | Renin secretion | http://www.genome.jp/kegg-bin/show_pathway?cfa04924 | ACE |
| | 0.0571 | 1 | 88 | 25.50379 | Protein digestion and absorption | http://www.genome.jp/kegg-bin/show_pathway?cfa04974 | ACE2 |
| | 0.0571 | 1 | 98 | 22.90136 | Chagas disease | http://www.genome.jp/kegg-bin/show_pathway?cfa05142 | ACE |
| | 0.0571 | 1 | 86 | 26.0969 | Small cell lung cancer | http://www.genome.jp/kegg-bin/show_pathway?cfa05222 | SECISBP2 |
| | 0.0571 | 1 | 88 | 25.50379 | Hypertrophic cardiomyopathy | http://www.genome.jp/kegg-bin/show_pathway?cfa05410 | ACE |
| | 0.0979 | 1 | 190 | 11.81228 | Diabetic cardiomyopathy | http://www.genome.jp/kegg-bin/show_pathway?cfa05415 | ACE |
| Sildenafil | 0.0000 | 3 | 76 | 88.59211 | Peroxisome | http://www.genome.jp/kegg-bin/show_pathway?cfa04146 | PEX5L, PEX5 PEX14 |
| | 0.0055 | 2 | 120 | 37.40556 | Purine metabolism | http://www.genome.jp/kegg-bin/show_pathway?cfa00230 | PDE5A, PDE11A |
| | 0.0447 | 1 | 45 | 49.87407 | Arginine and proline metabolism | http://www.genome.jp/kegg-bin/show_pathway?cfa00330 | CKMT2 |
| | 0.0447 | 3 | 1360 | 4.950735 | Metabolic pathways | http://www.genome.jp/kegg-bin/show_pathway?cfa01100 | CKMT2, PDE5A, PDE11A |
| | 0.0565 | 1 | 86 | 26.0969 | Morphine addiction | http://www.genome.jp/kegg-bin/show_pathway?cfa05032 | PDE11A |
| | 0.0565 | 1 | 86 | 26.0969 | Small cell lung cancer | http://www.genome.jp/kegg-bin/show_pathway?cfa05222 | SECISBP2 |
| | 0.0725 | 1 | 149 | 15.06264 | CGMP-PKG signaling pathway | http://www.genome.jp/kegg-bin/show_pathway?cfa04022 | PDE5A |
| | 0.0725 | 1 | 143 | 15.69464 | Cushing syndrome | http://www.genome.jp/kegg-bin/show_pathway?cfa04934 | PDE11A |
| | 0.1958 | 1 | 483 | 4.646653 | Pathways in cancer | http://www.genome.jp/kegg-bin/show_pathway?cfa05200 | SECISBP2 |

Pathway enrichment analysis was performed using the Kyoto Encyclopedia of Genes and Genomes (KEGG). Enriched pathways associated with each drug are presented with false discovery rate (FDR), gene counts, fold enrichment, and contributing genes.

disease processes rather than pharmacologic effects. In contrast, the MMVD C groups experienced substantial exposure to multiple cardiogenic medications, each of which may influence neurohormonal signaling, oxidative stress, renal hemodynamics, or mitochondrial metabolism [35,36]. These drug effects are therefore relevant contributors to the complex proteomic signatures observed in the MMVD C WOAZ and MMVD C WAZ groups. The differences in sildenafil and spironolactone usage between MMVD C WOAZ and MMVD C WAZ may also reflect variations in pulmonary hypertension severity or volume status, which could further modulate peptide expression [20,21]. However, the uniform administration of pimobendan and furosemide in both MMVD C subgroups ensures that treatment effects were generally consistent within disease stages.

Hematologic and biochemical values were generally stable across groups, indicating that most dogs did not exhibit systemic inflammation, severe anemia, or metabolic disturbances that might confound peptide expression patterns. The minor reductions in RBC count, hemoglobin, and hematocrit observed in the MMVD C WAZ and CKD stage 2 groups likely reflect early renal-associated anemia or hemodilution secondary to heart failure, but the magnitude of these changes was small and clinically insignificant. The biochemical results clearly differentiated azotemic from non-azotemic groups. The significant increases in creatinine and BUN in MMVD C WAZ and CKD stage 2 confirm renal involvement in these groups and validate their classification as cardiorenal or renal disease states. Importantly, albumin concentrations remained largely within normal limits, suggesting that neither protein-losing nephropathy nor marked malnutrition influenced peptide profiles. The absence of abnormalities in liver enzymes, glucose, leukogram parameters, and most red cell indices supports that systemic inflammatory or metabolic factors were not major contributors to the proteomic patterns identified. Thus, peptide alterations observed in MMVD C WAZ and CKD stage 2 likely stem from disease specific cardiorenal mechanisms such as decreased renal clearance, oxidative stress, neurohormonal activation, and tissue remodeling rather than generalized illness or unrelated systemic factors [37,38]

Radiographic and echocardiographic findings aligned well with expected pathophysiology across disease groups. The significant increases in VHS and VLAS in MMVD C WOAZ and MMVD C WAZ are consistent with progressive cardiac remodelling associated with chronic mitral regurgitation. These radiographic indicators confirmed left atrial and ventricular enlargement, which was further supported by echocardiographic increases in LVIDd, LVIDs, LA size, and LVIDdN. The absence of cardiac enlargement in both the healthy and CKD stage 2 groups indicates that renal disease alone did not contribute to measurable cardiac structural change in this cohort. This distinction reinforces that the peptide alterations identified in CKD stage 2 dogs are not attributable to occult cardiac disease. E- and -A wave elevations and the presence of MR Vmax and TR Vmax exclusively in the MMVD C groups confirm advanced diastolic loading conditions and valvular insufficiency [39]. These functional changes likely contribute to systemic neurohormonal activation, including RAAS stimulation [40], which is central in the development of cardiorenal interactions. In the present study, systolic function indices such as FS% did not differ significantly among groups. This observation is consistent with the pathophysiology of chronic mitral regurgitation in MMVD, in which cardiac dysfunction is primarily driven by chronic volume overload rather than intrinsic myocardial systolic failure. As a result, systolic indices such as FS% often remain within normal or even supranormal ranges due to compensatory hyperdynamic ventricular contraction and reduced effective afterload created by regurgitant flow into the left atrium [41]. Importantly, both preload and afterload conditions appeared relatively stable in the present cohort. The intravascular volume status of the enrolled dogs appeared physiologically stable, as total protein concentrations across all groups remained within normal reference ranges and showed no evidence of hemoconcentration suggestive of dehydration. In addition, systemic hypertension was excluded, with systolic blood pressure measurements remaining within the normal range. Because FS% is a load-dependent parameter influenced by changes in preload and afterload [42], the preservation of this index in MMVD stage C dogs in this study may reflect the expected hemodynamic characteristics of chronic mitral regurgitation rather than the absence of heart failure. These findings support the subsequent proteomic results by validating that MMVD C WOAZ and MMVD C WAZ groups represent distinct pathophysiological states involving cardiac remodelling, circulatory congestion, and increased neurohormonal burden factors that plausibly drive the peptide alterations identified through MALDI-TOF MS and nanoLC-MS/MS.

Urinalysis findings highlighted meaningful differences in renal function across groups, particularly decreased urine concentrating ability and mild UPCR elevations in dogs with advanced MMVD and CKD. While urine colour, clarity, and dipstick chemistry were not discriminatory, USG and UPCR showed potential as informative renal indicators in the context of cardiorenal involvement. USG was significantly reduced in all MMVD and CKD groups, reflecting impaired concentrating capacity likely related to early tubular dysfunction, reduced renal perfusion, or chronic hemodynamic alterations [43]. Importantly, dogs in the MMVD C WOAZ and MMVD C WAZ groups were receiving furosemide, which physiologically increases diuresis and further lowers USG, compounding the effect of cardiac disease. The MMVD C WAZ group

demonstrated the lowest USG values, consistent with azotemia and ultrasonographic evidence of reduced renal perfusion, supporting established CRS pathophysiologic mechanisms [44]. Although overt proteinuria was not present, UPCR was significantly higher in MMVD C WOAZ, MMVD C WAZ, and CKD stage 2 dogs, suggesting the possibility of subtle glomerular or tubular injury mediated by neurohormonal activation and hemodynamic stress [45]. Stable urine creatinine and urine protein values further suggest that UPCR may serve as a more sensitive early marker of renal compromise. Normal sediment and dipstick findings indicate that urinary tract inflammation or infection did not influence these patterns. Overall, the urinalysis data align closely with the proteomic findings, providing functional evidence of early renal impairment in MMVD C WAZ and CKD stage 2 dogs. The combination of reduced USG exacerbated by furosemide therapy in MMVD C groups and modest UPCR elevations supports the conclusion that subclinical renal changes in CRS are detectable both biochemically and at the peptide expression level.

Indirect systolic blood pressure measurements remained within the normotensive range in all enrolled dogs. Dogs with systemic hypertension (systolic blood pressure >140 mmHg) were excluded according to IRIS guidelines [14], minimizing the potential confounding effects of hypertension on renal or cardiovascular function. Hypertension is a well-recognized contributor to both renal injury and cardiac remodeling, particularly through increased afterload and glomerular hypertension. Therefore, the absence of systemic hypertension in this cohort suggests that the observed cardiac and renal alterations were primarily attributable to underlying MMVD or CKD rather than pressure-mediated vascular injury. These findings strengthen the interpretation that the proteomic alterations detected in this study are linked to cardiorenal pathophysiology rather than secondary effects of hypertension.

Peptide profiling was performed using MALDI-TOF MS and nanoLC-MS/MS. A key finding of this investigation was the identification of differentially expressed serum peptides potentially involved in the pathophysiological mechanisms underlying CRS, particularly in the MMVD C WAZ and CKD stage 2 groups and may aid in classification between groups and understanding of disease pathophysiology. MMVD staging was based on radiographic and echocardiographic criteria [13], while MMVD C WAZ and CKD stage 2 groups were classified using a combination of clinical parameters including BUN, serum creatinine levels, renal ultrasonographic findings (blood flow and morphology), systemic blood pressure, and urinalysis [25]. MMVD is the most common acquired cardiac disease in small-breed dogs, with the prevalence of this condition increasing with age [46]. Specifically, the prevalence of MMVD was found to be 23.9% in dogs aged 6–10 years and this rose to 30.3% in those aged 10–19 years [13]. However, in this study, the healthy group had a significantly younger average age compared to the other groups, since old animals are more likely to have other comorbidities, such as CKD or neoplasia. Nevertheless, all dogs in this study were of advanced age (>7 years) [47], and both male and female dogs were proportionately represented across all groups, suggesting that sex differences were unlikely to influence the study's findings. Consistent with previous proteomic studies in dogs [18,21], our group sizes (10–15 dogs) reflect practical constraints and case availability rather than a powered a priori statistical calculation. Although this sample size is appropriate for exploratory mass-spectrometry based biomarker studies, larger, prospectively powered cohorts will be needed to validate these candidate peptides. Both clinical and subclinical stages of MMVD have been identified as significant risk factors for the development and progression of kidney disease [8–12]. Consequently, close monitoring of renal function is essential during the management of cardiac disease. While conventional renal biomarkers such as serum creatinine and SDMA are routinely used, they have limitations in sensitivity and may be influenced by concurrent conditions [17].

The analysis using MALDI-TOF MS revealed a distinct PMFs separation of the healthy control group from all disease affected groups (MMVD and CKD) in the 2D PLS-DA plot. This finding confirms that chronic cardiac and renal pathologies induce significant and measurable changes in the serum peptidome, highlighting the potential of MALDI-TOF MS to serve as a rapid and reliable auxiliary screening tool for the presence of these systemic diseases in dogs. This result aligns with earlier MALDI-TOF MS studies in veterinary cardiology, which similarly observed discrete clustering and distinct peptide barcodes between healthy and diseased cohorts [21]. However, the observed close clustering among the MMVD subgroups and the CKD stage 2 group suggests a considerable overlap in their low molecular weight peptidomic signatures,

indicating that the peptides detected may represent shared systemic pathophysiological processes such as chronic inflammation, compromised hemostasis, or common degradation pathways related to multiple organ dysfunction (e.g., cardiorenal syndrome) [48]. This approach, coupled with the rigorous sample preparation and analysis confirmed by the internal homogeneity of the pooled samples, provides a strong foundation for future validation studies necessary to translate these promising PMFs or peptide barcodes into precise and clinically actionable biomarkers for the differential diagnosis and monitoring of canine MMVD and concurrent CKD.

All peptides and proteins were identified from each of the pooled serum samples by using nanoLC-MS/MS, successfully delineating distinct molecular signatures across the five experimental groups: Healthy, MMVD B1, MMVD C WOAZ, MMVD C WAZ, and CKD Stage 2. The identification of 100 statistically significant proteins based on ANOVA ($p < 0.05$) points to a core set of proteins intimately involved in the pathophysiology of these chronic conditions. A critical finding is the presence of a large subset of 510 common proteins and specifically, six shared statistically significant proteins (PEX14, GLRX2, CKMT2, SECISBP2, LRRCT domain-containing protein, and HINFP) between the MMVD C WAZ and CKD Stage 2 groups. The identification of six shared statistically significant proteins between the MMVD C WAZ and CKD Stage 2 groups provides a molecular basis for understanding the progression of cardiorenal dysfunction. These proteins serve as potential prognostic biomarkers and may act as indicators of treatment efficacy in dogs receiving pimobendan, furosemide, spironolactone, benazepril, and sildenafil [13]. The drug-protein interaction network revealed that these significant proteins are deeply embedded in pathways that appear consistently across multiple drug interactions, highlighting their systemic role in cardiac and kidney health.

In our study, peroxisome and arginine and proline metabolism were found to be associated with all five drugs used in this study (pimobendan, furosemide, spironolactone, benazepril, and sildenafil). The systemic nature of CRS is increasingly understood as a convergence of metabolic failure and oxidative stress. Central to this process is the role of peroxisomes. Peroxisomes are essential for the β-oxidation of very-long-chain fatty acids, substrates that mitochondria cannot efficiently metabolize [49]. Impaired peroxisomal function disrupts this lipid oxidation pathway, a metabolic hallmark documented in mammalian heart failure models and dogs with failing myocardium [50]. Furthermore, peroxisomes regulate cellular redox balance by producing and scavenging reactive oxygen and nitrogen species (ROS) [51,52]. The clinical importance of these metabolic functions is underscored by the protective role of peroxisomal enzymes in cardiovascular and renal [53,54]. Specifically, the overexpression of catalase (CAT), a primary peroxisomal antioxidant, facilitates significant cardioprotection by reducing oxidized lipid-induced cytotoxicity and preventing pathological mechanical alterations in vascular smooth muscle cells. Furthermore, CAT overexpression in renal tubular cells provides essential renoprotection, as demonstrated in various mouse models where it mitigates oxidative stress to prevent the development of hypertension, albuminuria, and tubulointerstitial fibrosis [55].

Building upon this metabolic foundation, our proteomic analysis identified a panel of significant proteins, including PEX14, GLRX2, CKMT2, and SECISBP2 that collectively point toward a unifying mechanism of peroxisomal and mitochondrial failure. The identification of PEX14 as a shared, statistically significant protein in both MMVD with azotemia and CKD stage 2 groups is critical. PEX14 acts as the primary docking site for peroxisomal protein import [56]. Specifically, dysregulation of PEX14 disrupts the import of antioxidant enzymes like CAT, which normally provides cardioprotection and renoprotection by mitigating oxidative stress, preventing albuminuria, and reducing tubulointerstitial fibrosis. Notably, while our analysis uniquely associated PEX14 with pimobendan using the Stitch (EMBL) database, its broader role suggests that the lack of efficient protein import promotes hydrogen peroxide ($H_2O_2$) mediated injury and dysregulated lipid metabolism, triggering pro-fibrotic signaling in both the heart and kidneys [57,58]. Complementing this peroxisomal failure is the significant detection of GLRX2 and CKMT2, which reflect severe mitochondrial and energetic compromise [59,60]. As a mitochondrial redox enzyme, GLRX2 is vital for protecting mitochondria from oxidative damage [59,61]. Its insufficiency leads to the accumulation of S-glutathionylation on metabolic enzymes, reducing ATP synthesis and establishing a self-perpetuating cycle of systemic injury characteristic of CRS [36,38,62]. Concurrently, the sarcomeric isoform CKMT2 is essential for ATP regeneration through the phosphocreatine shuttle [58]. Vulnerability to ROS leads to the oxidation of

CKMT2, suppressing its enzymatic activity and causing mitochondrial overload and inefficient contractile performance [63,64]. This energetic failure, common to both advanced MMVD and CKD, facilitates the leakage of peptide fragments into the bloodstream, a pattern observed in the proteomic profiles of our disease cohorts. Finally, the identification of SECISBP2 underscores a systemic failure in cellular redox capacity [65]. SECISBP2 is the key determinant for synthesizing antioxidant selenoproteins, such as glutathione peroxidases and thioredoxin reductases [66]. The oxidative stress and inflammation hallmarks of heart disease and CKD likely impair SECISBP2 activity, suppressing downstream antioxidant defenses [67,68]. In the myocardium, this leads to mitochondrial injury and sarcomeric destabilization, while in the kidney, it contributes to tubular oxidative stress and inflammation [69]. Together, the alterations in PEX14, GLRX2, CKMT2, and SECISBP2 demonstrate that peroxisomal and mitochondrial dysfunctions are convergent drivers of metabolic derangement and structural injury in the broader context of canine cardiorenal syndrome.

Arginine and proline metabolism represents a central metabolic hub where imbalances directly contribute to the pathogenesis of both cardiovascular and renal diseases. Reduced nitric oxide (NO) synthesis from L-arginine often caused by endothelial dysfunction promotes atherosclerotic progression and the deterioration of renal function [70]. Furthermore, impaired proline catabolism via proline dehydrogenase reduces mitochondrial energy supply and ATP synthesis, establishing a link between metabolic shifting and myocardial remodeling [71]. Within the kidney, these pathways govern redox balance and apoptosis regulation, where L-proline levels specifically influence cellular survival and protect against fibrotic tissue damage [72]. Ultimately, the integration of these amino acid pathways underscores their role as both biomarkers of disease progression and therapeutic targets for ameliorating cardiorenal syndrome [70]. The identification of shared alterations in PEX14 and the arginine and proline metabolism pathway suggests an integrated peroxisomal metabolic axis in the pathogenesis of canine cardiorenal syndrome. PEX14 is a critical regulator of peroxisomal biogenesis and protein docking; its dysregulation, potentially influenced by phosphorylation changes, impairs organelle functions such as fatty acid β-oxidation and ROS scavenging [49,56]. This peroxisomal failure likely disrupts the biosynthesis of protective metabolites like proline and putrescine, which are essential for osmotic regulation, energy supply, and mitigating oxidative injury in failing cardiac and renal tissues [73]. Consequently, the convergence of PEX14 mediated peroxisomal dysfunction and impaired amino acid catabolism establishes a unifying mechanism for the energetic and structural remodeling observed in cardiorenal deterioration. The convergence of GLRX2 and the arginine-proline metabolic axis reveals a sophisticated redox-shielding mechanism that is likely compromised in canine cardiorenal disease. GLRX2 serves as a critical mitochondrial sensor, utilizing its iron-sulfur center to release active monomers during oxidative stress, which then catalyze protein deglutathionylation to maintain enzymatic integrity [74]. This antioxidant response is bolstered by L-arginine, which activates the Nrf2-Keap1 pathway to upregulate essential glutathione-dependent enzymes, and by proline metabolism, which facilitates a proadaptive stress resistance by enhancing catalase and glutaredoxin activities. In the context of the cardiorenal syndrome, the failure of this integrated system potentially driven by the previously discussed peroxisomal PEX14 dysfunction impairs the mitochondrial matrix's buffering capacity [75]. When GLRX2-mediated deglutathionylation and the Nrf2-driven antioxidant supply fail, the resulting unchecked oxidative stress and energetic decline accelerate the structural remodeling and fibrosis observed in both heart and kidney tissues [62]. The association between the arginine-proline metabolism pathway and the creatine kinase system (CKMT2/CKMT1) highlights a critical metabolic axis for cellular energy buffering that is likely disrupted in cardiorenal disease. Arginine serves as a direct biosynthetic precursor for creatine, and our findings suggest that intracellular arginine pools dictate the capacity for creatine synthesis, which is essential for the phosphocreatine shuttle governed by mitochondrial CKMT2 [60]. In the context of the cardiorenal syndrome, the energy failure observed in failing myocardium and renal tubules may stem from a breakdown in this axis; specifically, impaired arginine availability limits the synthesis of creatine, thereby compromising the ATP-buffering capacity of CKMT2 [76]. While this metabolic shift is driven by mitochondrial and cytosolic demand, it may be further exacerbated by the previously discussed peroxisomal dysfunction, as peroxisomes regulate the upstream redox environment and amino acid catabolism necessary to maintain these arginine-driven energetic pathways. Although direct reports linking

SECISBP2 specifically to arginine or proline metabolism are scarce, these pathways are functionally interdependent through the thiol-redox system [66]. This protein represents a critical regulatory layer that ensures the functional expression of the antioxidant systems previously discussed, such as the glutathione-dependent pathways linked to GLRX2.

Moreover, the identification of altered abundance in PEX14, GLRX2, CKMT2, and SECISBP2 provides a high-resolution molecular snapshot of the cellular stress encountered in dogs with advanced MMVD and CKD. We propose that these proteins serve as systemic markers of the strain caused by sustained RAAS activation and the resultant oxidative burden, which persists despite standard pharmacological intervention. The standard therapy consisting of pimobendan, furosemide, spironolactone, and benazepril, often supplemented with sildenafil aims to control congestion and modulate neurohormonal signaling. However, the persistence of these protein markers suggests a deep-seated disruption of organellar redox homeostasis. Furosemide acts at the sodium potassium chloride cotransporter 2 transporter to promote natriuresis, but its use simultaneously triggers renin release, leading to increased angiotensin II and aldosterone production [77]. This signaling stimulates nicotinamide adenine dinucleotide phosphate oxidase activity and mitochondrial ROS generation [78]. While benazepril and spironolactone attempt to mitigate this by limiting Angiotensin II conversion and blocking mineralocorticoid receptors, respectively, the chronic RAAS signaling continues to drive excessive ROS via mitochondrial pathways, contributing to myocardial remodeling and renal tubular injury [79].

Our proteomic findings suggest that this oxidative environment leads to peroxisomal and mitochondrial failure. The dysregulation of PEX14 is critical, as it impairs the import of vital antioxidant enzymes like catalase, thereby increasing $H_2O_2$-mediated injury in both the heart and kidneys [80]. Concurrently, changes in GLRX2 and CKMT2 reflect a compensatory but insufficient response to energetic compromise and mitochondrial redox imbalance [74,81]. The identified organellar dysfunction characterized by peroxisomal and mitochondrial failure are intrinsically linked to the dysregulation of arginine and proline metabolism, a central metabolic hub where biochemical supply and demand converge to accelerate cardiorenal deterioration. Arginine serves as the essential precursor for NO, a critical signaling molecule for vasodilation and renoprotection; however, the chronic oxidative stress environment of CRS reduces NO bioavailability, a deficit that pharmacological interventions like sildenafil attempt to restore by enhancing the cGMP–PKG signaling axis [82]. Ultimately, the convergence of RAAS driven oxidative stress and the relative suppression of this protective cGMP–PKG axis facilitates a toxic metabolic environment that reinforces peroxisomal and mitochondrial dysfunction. This environment shifts systemic metabolic pathways, particularly arginine and proline metabolism, toward maladaptive states: the depletion of arginine pools limits vasodilatory capacity, while dysregulated proline metabolism promotes collagen synthesis and fibrotic remodeling. These metabolic shifts do not occur in isolation but act as a self-perpetuating cycle, where impaired organellar energy production and amino acid imbalances contribute to the parallel cardiac and renal structural deterioration observed in canine cardiorenal syndrome.

This study has several limitations that should be considered when interpreting the results. First limitation is that systolic function was evaluated primarily using FS%, which is a load-dependent parameter influenced by changes in preload and afterload. Therefore, FS% may not fully reflect intrinsic myocardial contractility in this disease context. Future studies may benefit from incorporating less load-dependent indices, such as tissue Doppler imaging, speckle-tracking echocardiography–derived myocardial strain, or the Tei index, which may provide a more sensitive assessment of ventricular function in canine cardiorenal syndrome. Second, the sample size (10–15 dogs per group) was relatively small, reflecting the exploratory nature of the study and case availability. While sufficient for identifying candidate proteins, larger prospective cohorts are needed for validation. Third, the group classification relied on clinical parameters and international guidelines [13,14]; however, individual biological variability and subclinical comorbidities may still influence peptidomic results. The lack of significant changes in Stage B1 and non-azotemic MMVD groups suggests that the identified mitochondrial and peroxisomal dysfunctions are specific to the more severe, systemic phase of cardiorenal syndrome. Fourth, the use of pooled samples for nanoLC-MS/MS was necessary to identify core protein differences between disease states, but it limits the assessment of individual variation within groups. Fifth, since this study focused specifically on Type 2 cardiorenal

syndrome with a small sample size, these peptidomic signatures may not apply to all CRS types, such as Type 4. Additionally, although these profiles were observed in azotemic cases, they were absent in MMVD cases without azotemia (WOAZ), suggesting that these markers reflect later stages of the cardiorenal continuum rather than early subclinical detection. These signatures provide valuable insights into the molecular remodeling associated with Type 2 CRS, but their generalizability remains to be established. Furthermore, we acknowledge the limitation that our data demonstrate an association between current therapeutic protocols and the maintenance of certain metabolic pathways, rather than causal links. As this study involved client-owned animals requiring standard-of-care, we cannot definitively rule out whether some peptide fragments are exacerbated by the treatments themselves (e.g., drug-induced RAAS activation) in the absence of a treatment-naïve control group. This clarification is included to ensure a balanced interpretation of the molecular changes observed under standard medical therapy. The absence of these datasets limits the generalizability of our findings across the entire cardiorenal syndrome spectrum, and future research incorporating treatment-naïve cases and varying CRS subtypes is warranted to validate these proteomic signatures. To mitigate this, our study design utilized MALDI-TOF MS and nLC-MS/MS as complementary steps in a discovery pipeline. MALDI-TOF MS was employed as a high-throughput screening tool to evaluate the global peptide "fingerprint" and to verify the homogeneity of individual samples within each clinical group. This ensured that the subsequent nLC-MS/MS analysis was performed on representative pooled samples. While the PMF peaks provide a rapid means of group classification based on mass distribution, our primary objective for the nLC-MS/MS analysis was to perform a deep-dive into the peptidomic landscape to identify specific biomarkers and their associated pathways (e.g., mitochondrial and peroxisomal failure). Although we did not perform a direct one-to-one assignment of MALDI peaks to peptide sequences, the nLC-MS/MS results provide the necessary biological identity to the peptidomic alterations observed during the initial screening. Finally, MALDI-TOF MS remains an investigational tool, and the resulting peptide mass fingerprints, as well as the low-molecular-weight peptides identified, can be difficult to confirm using conventional laboratory assays, requiring specialized equipment for future clinical application.

## Conclusion

Our study demonstrates that canine cardiovascular-renal dysfunction originating from cardiovascular disease (CvRDh) is characterized by distinct serum peptide and protein alterations reflecting a systemic failure of cellular redox and energetic homeostasis. The identification of PEX14, GLRX2, CKMT2, and SECISBP2 as key proteomic signatures highlights a common association of peroxisomal and mitochondrial dysfunction, driven by chronic RAAS activation and oxidative stress. These molecular changes persist despite standard medical therapy, indicating that current therapies might not fully address the fundamental organellar metabolic failure. Our results suggest that proteomic profiling and PMFs have significant potential as additional tools for tracking the progression and risk assessment of advanced kidney disease in dogs with heart failure. These peptidomic characteristics could not be applicable to other CRS types, though, because this study only examined Type 2 cardiorenal syndrome with a small sample size. Furthermore, these profiles were lacking in MMVD cases without azotemia (WOAZ), although being seen in azotemic cases, indicating that these markers represent later stages of the cardiorenal continuum. These signatures provide valuable insights into the molecular remodeling associated with Type 2 CRS, potentially leading to more targeted and effective treatment plans for affected dogs. Future studies should investigate these Type 2 CRS groups to further validate the role of mitochondrial and peroxisomal biomarkers across all CRS subtypes.

## Supporting information

**S1 Table. Medication of the enrolled dog.** Abbreviations: Healthy, healthy control; MMVD B1, myxomatous mitral valve disease stage B1; MMVD C WOAZ, MMVD stage C without azotemia; MMVD C WAZ, MMVD stage C with azotemia; CKD stage 2, chronic kidney disease at IRIS stage 2. Data are presented as proportions (percentages).
(DOCX)

**S2 Table. Complete blood count and blood chemistry profiles of dogs in the experimental groups.** Abbreviations: RBC, red blood cell; Hb, hemoglobin; Hct, hematocrit; MCV, mean corpuscular volume; MCH, mean corpuscular hemoglobin; MCHC, mean corpuscular hemoglobin concentration; WBC, white blood cell; ALT, alanine aminotransferase; ALP, alkaline phosphatase; BUN, blood urea nitrogen; Healthy, healthy control; MMVD B1, myxomatous mitral valve disease stage B1; MMVD C WOAZ, MMVD stage C without azotemia; MMVD C WAZ, MMVD stage C with azotemia; CKD stage 2, chronic kidney disease at IRIS stage 2. Statistical differences among groups were assessed using the Kruskal–Wallis test followed by the Mann–Whitney U test. Superscript letters indicate statistically significant differences between groups (P<0.05). Data are presented as medians with interquartile ranges. Reference intervals were obtained from the Clinical Laboratory of Prasu Arthorn Veterinary Teaching Hospital.
(DOCX)

**S3 Table. Thoracic radiographic findings and echocardiographic indices of the enrolled dog.** Echocardiographic indices were normalized using Cornell's allometric scaling method. Abbreviations: VHS, vertebral heart score; VLAS, vertebral left atrial score; IVSd, interventricular septal thickness at end-diastole; LVIDd, left ventricular internal diameter at end-diastole; LVPWd, left ventricular posterior wall thickness at end-diastole; IVSs, interventricular septal thickness at end-systole; LVIDs, left ventricular internal diameter at end-systole; LVPWs, left ventricular posterior wall thickness at end-systole; LA, left atrium; AO, aorta; LA:AO, left atrium-to-aorta ratio; FS, fractional shortening; Healthy, healthy control; MMVD B1, myxomatous mitral valve disease stage B1; MMVD C WOAZ, MMVD stage C without azotemia; MMVD C WAZ, MMVD stage C with azotemia; CKD stage 2, chronic kidney disease at IRIS stage 2. Statistical differences among groups were evaluated using the Kruskal–Wallis test, with post hoc pairwise comparisons performed using the Mann–Whitney U test. Superscript letters indicate statistically significant differences between groups for each variable (P<0.05); groups sharing the same letter are significantly different. Data are presented as medians with interquartile ranges.
(DOCX)

**S4 Table. Urinalysis indices of the enrolled dog.** Abbreviations: Healthy, healthy control; MMVD B1, myxomatous mitral valve disease stage B1; MMVD C WOAZ, MMVD stage C without azotemia; MMVD C WAZ, MMVD stage C with azotemia; CKD stage 2, chronic kidney disease at IRIS stage 2. Statistical differences among groups were evaluated using the Kruskal–Wallis test with post hoc pairwise comparisons performed using the Mann–Whitney U test. Categorical data were analyzed using the chi-square test. Superscript letters indicate statistically significant differences between groups for each variable (P<0.05); groups sharing the same letter are significantly different. Data are presented as medians with interquartile ranges.
(DOCX)

**S5 Table. A quantitative proteomic profile of serum proteins across Healthy, MMVD, MMVD with and without azotemia, and CKD stage II dogs.** Abbreviations: Healthy, healthy control; MMVD B1, myxomatous mitral valve disease stage B1; MMVD C WOAZ, MMVD stage C without azotemia; MMVD C WAZ, MMVD stage C with azotemia; CKD stage 2, chronic kidney disease at IRIS stage 2.
(XLSX)

**S6 Table. Statistical analysis of differentially expressed serum proteins among Healthy, MMVD, MMVD with and without azotemia, and CKD stage II dogs.** Abbreviations: Healthy, healthy control; MMVD B1, myxomatous mitral valve disease stage B1; MMVD C WOAZ, MMVD stage C without azotemia; MMVD C WAZ, MMVD stage C with azotemia; CKD stage 2, chronic kidney disease at IRIS stage 2. Statistical analysis was performed to evaluate differentially expressed serum proteins among Healthy, MMVD B1, MMVD C with and without azotemia, and CKD stage 2 dogs.
(XLSX)

**S7 Table. Functional pathway enrichment analysis of differentially expressed serum proteins in canine cardiorenal syndrome.** Abbreviations: Healthy, healthy control; MMVD B1, myxomatous mitral valve disease stage B1; MMVD C WOAZ, MMVD stage C without azotemia; MMVD C WAZ, MMVD stage C with azotemia; CKD stage 2, chronic kidney disease at IRIS stage 2.
(XLSX)

**S8 Table. Predicted upstream regulators and disease associations of differentially expressed serum proteins in canine cardiorenal syndrome.** Gene Ontology enrichment analysis was performed to identify significantly overrepresented biological processes associated with the differentially expressed proteins. Enriched GO terms are presented with false discovery rate (FDR), number of input genes (nGenes), total genes annotated to each pathway (Pathway Genes), fold enrichment, and contributing genes. URLs provide direct access to corresponding Gene Ontology pathway annotations.
(XLSX)

**S9 Table. Key serum protein biomarkers associated with disease progression in canine cardiorenal syndrome.** Gene Ontology enrichment analysis was conducted to identify significantly enriched molecular functions. Results are reported with false discovery rate (FDR), number of associated genes (nGenes), total genes annotated to each term (Pathway Genes), fold enrichment, and the corresponding gene list. Hyperlinks direct to the detailed Gene Ontology term descriptions.
(XLSX)

## Author contributions

**Conceptualization:** Poonavit Pichayapaiboon, Sekkarin Ploypetch, Walasinee Sakcamduang, Ruangrat Buddhirongawatr.

**Data curation:** Poonavit Pichayapaiboon, Sekkarin Ploypetch, Walasinee Sakcamduang, Sittiruk Roytrakul, Wachira Trakoolchaisri, Ruangrat Buddhirongawatr.

**Formal analysis:** Poonavit Pichayapaiboon, Sekkarin Ploypetch, Walasinee Sakcamduang, Sittiruk Roytrakul, Janthima Jaresitthikunchai, Narumon Phaonakrop, Ruangrat Buddhirongawatr.

**Funding acquisition:** Poonavit Pichayapaiboon, Ruangrat Buddhirongawatr.

**Investigation:** Poonavit Pichayapaiboon, Sekkarin Ploypetch, Walasinee Sakcamduang, Wachira Trakoolchaisri.

**Methodology:** Poonavit Pichayapaiboon, Sekkarin Ploypetch, Walasinee Sakcamduang, Sittiruk Roytrakul, Janthima Jaresitthikunchai, Narumon Phaonakrop.

**Project administration:** Poonavit Pichayapaiboon, Ruangrat Buddhirongawatr.

**Supervision:** Poonavit Pichayapaiboon, Sekkarin Ploypetch, Walasinee Sakcamduang, Ruangrat Buddhirongawatr.

**Validation:** Poonavit Pichayapaiboon, Sekkarin Ploypetch, Walasinee Sakcamduang, Sittiruk Roytrakul, Ruangrat Buddhirongawatr.

**Visualization:** Poonavit Pichayapaiboon, Sekkarin Ploypetch.

**Writing – original draft:** Poonavit Pichayapaiboon, Sekkarin Ploypetch, Walasinee Sakcamduang, Ruangrat Buddhirongawatr.

**Writing – review & editing:** Poonavit Pichayapaiboon, Sekkarin Ploypetch, Walasinee Sakcamduang, Sittiruk Roytrakul, Janthima Jaresitthikunchai, Narumon Phaonakrop, Wachira Trakoolchaisri, Ruangrat Buddhirongawatr.

## Acknowledgments

The authors wish to express their sincere gratitude to the Faculty of Veterinary Science, Mahidol University (Thailand), and the dedicated staff at Prasu Arthorn Veterinary Teaching Hospital, Faculty of Veterinary Science, Mahidol University, Nakhon Pathom, Thailand, for their valuable support and assistance throughout this study. Special thanks are also extended to all the dog owners who generously participated in the research.

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
