## [Decision Letter · Decision Letter 0]

10 Mar 2026

PONE-D-25-68574Serum Peptidomic Profiling and Peptide Mass Fingerprinting Reveal Signatures Associated with Peroxisomal and Mitochondrial Pathways in Canine Cardiorenal SyndromePLOS One

Dear Dr. pichayapaiboon,

Thank you for submitting your manuscript to PLOS ONE. After careful consideration, we feel that it has merit but does not fully meet PLOS ONE’s publication criteria as it currently stands. Therefore, we invite you to submit a revised version of the manuscript that addresses the points raised during the review process.

Please review the comments from the two expert reviewers below and revise your manuscript to address their suggestions, providing a point-by-point response to each comment upon resubmission.

We look forward to receiving your revised manuscript.

Kind regards,

Sarah Jose, Ph.D.

Staff Editor

PLOS One

Journal Requirements:

2. To comply with PLOS One submissions requirements, in your Methods section, please provide additional information regarding the experiments involving animals and ensure you have included details on (1) methods of sacrifice, (2) methods of anesthesia and/or analgesia, and (3) efforts to alleviate suffering.

This research project is supported by the Faculty of Veterinary Science, Mahidol University.

6. Please upload a new copy of Figure 3 as the detail is not clear. Please follow the link for more information:  https://journals.plos.org/plosone/s/figures

Reviewers' comments:

Reviewer's Responses to Questions

**Comments to the Author**

1. Is the manuscript technically sound, and do the data support the conclusions?

Reviewer #1: Yes

Reviewer #2: Partly

2. Has the statistical analysis been performed appropriately and rigorously? 

Reviewer #1: No

Reviewer #2: Yes

3. Have the authors made all data underlying the findings in their manuscript fully available?

Reviewer #1: Yes

Reviewer #2: No

4. Is the manuscript presented in an intelligible fashion and written in standard English?

Reviewer #1: Yes

Reviewer #2: Yes

5. Review Comments to the Author

Reviewer #1: To differentiate disease stages of cardiorenal syndrome in dogs and identify key molecules involved in disease pathophysiology, the authors, in their manuscript entitled "Serum Peptidomic Profiling and Peptide Mass Fingerprinting Reveal Signatures Associated with Peroxisomal and Mitochondrial Pathways in Canine Cardiorenal Syndrome," performed PMF and nLC-MS/MS analysis of peptide fractions (after 10 kDa ultrafiltration of serum samples). Small-breed dogs (64 dogs of various breeds, sexes, and ages; Table 1) were classified into five groups: Healthy (N=15), MMVD (myxomatous mitral valve degeneration) Stage B1 (N=10), MMVD Stage C without azotaemia (N=15), MMVD Stage C with azotaemia (N=13), and CKD (chronic kidney disease) IRIS Stage 2 (N=11). Dogs in the MMVD Stage C without azotaemia and MMVD Stage C with azotaemia groups were receiving different therapies (medications) as detailed in Table S1. Prior to MS-based analyses, comprehensive diagnostics were performed, including haematology and blood chemistry, urine analysis, blood pressure measurement, thoracic radiography, echocardiography, and abdominal ultrasonography.

PMF by MALDI-TOF MS revealed separation of the healthy control group from all disease-affected groups (MMVD and CKD), highlighting the potential of MALDI-TOF MS as a rapid screening tool. However, close clustering was observed among the MMVD subgroups and the CKD stage 2 group.

Peptidome analysis identified 100 statistically significant proteins, of which six (PEX14, GLRX2, CKMT2, SECISBP2, LRRCT domain-containing protein, and HINFP) were selected as disease markers involved in peroxisomal and mitochondrial failure resulting from chronic RAAS activation and oxidative stress. Gene ontology and pathway analysis of the identified proteins, as well as protein–protein and protein–drug interactions of candidate peptides and drugs, were conducted. The authors noted that the interactions of these six significant proteins with cardiogenic drugs investigated in this study are linked to the pathophysiology of CRS, particularly involving arginine and proline metabolism and peroxisome pathways.

The authors provided the ethical statement details for this clinical study (approval number MUVS2023-03-20).

The manuscript is well structured and follows the author's instructions. The abstract is clearly written, providing the purpose of the research, methods used, main results, and major conclusions. The methods are well chosen to address the research question, appropriately applied to obtain statistically sound conclusions, and well described, although some important details are missing. The conclusions are based on the results obtained and are supported by the scientific literature cited in the manuscript. The supplementary files are very informative. The proteomic data are deposited in the PRIDE repository.

The authors also highlight the limitations and strengths of their study. The limitations mainly relate to sample size and animal age; however, this is understandable, as this is a clinical study and it is not easy to have ideal groups. Furthermore, pooled samples were used for peptidomic/proteomic analysis which could be a major issue if no replicates were analyzed. No significant self-citation is observed; hovewer the self-citations (maximum 5) demonstrate the authors’ expertise in the field.

The manuscript would be significant contribution to multiple fields such as biomedicine and animal welfare. However, there are some parts needing corrections and/or more details. For this reason, I recommend this manuscript to be accepted after major revision.

Major:

1. For nLC-MS/MS analysis, please provide details related to related to sample preparation (if also up to 10 kDa fractions were used), sample pooling and No. of technical replicates. From the Supplementary data it could be seen samples were analysed in triplicate. Please add this, otherwise the quantification results would not be confident if only one sample per group was used for statistical analysis.

Furthermore, please clarify how many unique peptides were used for protein identification (line 270).

2. In the Table 2 the authors presented significant proteins, however GO data for several proteins is missing. Could you please update the table using available databases such as:

https://www.genecards.org/cgi-bin/carddisp.pl?gene=GLRX2&keywords=GLRX2

https://maayanlab.cloud/Harmonizome/gene/SECISBP2

https://www.ebi.ac.uk/interpro/protein/UniProt/A0A8D2EGW1/

or similar. Since data for canine genes/proteins are often missing, please use human gene data (which is acceptable). I believe this would improve the quality of your manuscript.

3. It would be interesting to comment if some distinctive peaks (in terms of m/z) obtained by MALDI-TOF MS (PMF) were detected and/or identified by nLC-MS/MS. It would provide more strength to PMS results and its further clinical application.

Minor changes:

Line 32. Please provide full name for CKD.

Table 3. – Please change the title: Protein-drug pathway enrichment analysis performed using the Kyoto Encyclopedia of Genes and Genomes (KEGG). Furthermore, in FC enrichment and Enrichment FDR, please have in mind the number of decimal places, they should be the same within the column. I hope the text within the column names will be formatted/resized during paper publishing (e.g. Genes number, etc.).

Supplementary table 2 - Please correct: b Statistically significant difference compared Statistically significant difference compared between groups (P < 0.05).

Reviewer #2: First of all, I would like to appreciate the authors for selecting an important subject and for their extensive data analysis, well-designed experimental methods, and factual and comprehensive conclusions. As an academician and scientist in veterinary medicine, I would like to comment on certain methodological and sampling aspects that remain unaddressed, along with a few differences of opinion regarding some of the conclusions drawn.

1. The data only partially support the conclusions

a) The authors conclude that proteomic profiling and peptide mass fingerprints (PMFs) offer significant potential as auxiliary tools for the early detection of kidney disease in cardiorenal syndrome (CRS). However, in the present study, these proteomic profiling and PMF approaches identified renal pathology only in the two groups where conventional markers such as creatinine were already elevated. They did not identify renal pathology in MMVD non-azotemic dogs, where early renal disease initiation is still possible according to the CRS concept. Therefore, data demonstrating the renal marker potential of these proteomic signatures in subclinical renal disease associated with CRS would strengthen this conclusion.

b) The authors also state that canine cardiorenal syndrome is characterized by distinct serum peptide and protein alterations reflecting systemic failure of cellular redox and energetic homeostasis. However, the study evaluates only 13 cases belonging to Type 2 CRS (chronic heart disease with kidney injury; MMVD with azotemia) and does not prove cardiac pathology in chronic kidney disease cases (Type 4 CRS). It may therefore be inappropriate to generalize this pattern to the broader CRS condition when only one of the five CRS subtypes was studied with a limited sample size. It may be more appropriate to limit the conclusion to cardiovascular-renal dysfunction originating from cardiovascular disease (CvRDh). If the authors have data from other CRS groups, including them would help substantiate this broader conclusion. Additionally, the study does not establish peptide profiles indicating peroxisomal and mitochondrial dysfunction related to myocardial remodeling in MMVD without azotemia (WOAZ) cases.

c) The study further concludes that the identified molecular changes persist despite standard medical therapy, suggesting that current treatments may not fully mitigate underlying organellar metabolic failure. However, this conclusion lacks supporting comparative data from untreated dogs, where proteomic changes might differ. Moreover, as the authors themselves mention, certain treatment protocols may augment azotemia, renal pathology, renin release, and subsequent RAAS activation. This could potentially exaggerate cardiac and/or renal peptide fragments in treated groups compared with untreated animals. Therefore, establishing this conclusion would require inclusion of treatment-naïve cases for comparison.

I agree with the authors’ concluding statement regarding the potential role of these findings in risk stratification, as the study indeed provides valuable insights into key molecular components involved in myocardial and renal remodeling in CRS. However, similar expected proteomic data reflecting myocardial remodeling in subclinical or non-azotemic MMVD cases are lacking in the present study.

2. Availability of underlying data supporting the findings

a) As mentioned above, data regarding the proteomic profile in cardiovascular disease/dysfunction secondary to renal disease and proteomic profile and PMFs in untreated MMVD Stage C dogs are not available in the manuscript. Inclusion of these datasets would help reinforce the conclusions.

b) MMVD Stage C constituted an important sampling group in the present study. However, the Materials and Methods section does not clearly describe the criteria used for selecting this group. Details regarding the diagnostic criteria for MMVD Stage C, duration of illness, and treatment status should be incorporated. In addition, the authors should substantiate the reported normal systolic function findings in MMVD dogs that are described as having heart failure.

c) The proteomic profiles and PMFs identified were associated with metabolic dysfunctions relevant to renal pathology (CKD) and cardiorenal pathology (MMVD with azotemia). However, no data are presented to demonstrate that these markers are associated with isolated cardiac pathology, such as in MMVD Stage B1 or MMVD without azotemia. It is therefore possible that these proteomic signatures are present primarily in advanced renal (could be primary or secondary as in CRS) disease states, rather than in isolated cardiac disease or early renal pathology.

d) Although blood pressure was mentioned in the Materials and Methods section as a significantly differential parameter, the study does not present the blood pressure data for any of the study groups.

6. PLOS authors have the option to publish the peer review history of their article (what does this mean?). If published, this will include your full peer review and any attached files.

Reviewer #1: No

Reviewer #2: **Yes:**Dr. Umesh C.G.

---

## [Author Response · Author response to Decision Letter 1]

25 Mar 2026

Thank you for the thorough and helpful review of our paper. We appreciate the opportunity to improve our work for publication in PLoS ONE. We have addressed all the concerns raised during the review, especially regarding the statistical replicates, animal ethics descriptions, and the scope of our conclusions.

---

## [Decision Letter · Decision Letter 1]

13 Apr 2026

Serum peptidomic profiling and peptide mass fingerprinting reveal signatures associated with peroxisomal and mitochondrial pathways in MMVD-associated cardiorenal syndrome in dogs

PONE-D-25-68574R1

Dear Dr. Buddhirongawatr,

We’re pleased to inform you that your manuscript has been judged scientifically suitable for publication and will be formally accepted for publication once it meets all outstanding technical requirements.

Kind regards,

Vincenzo Lionetti, M.D., PhD

Academic Editor

PLOS One

Additional Editor Comments (optional):

Reviewers' comments:

Reviewer's Responses to Questions

**Comments to the Author**

1. If the authors have adequately addressed your comments raised in a previous round of review and you feel that this manuscript is now acceptable for publication, you may indicate that here to bypass the “Comments to the Author” section, enter your conflict of interest statement in the “Confidential to Editor” section, and submit your "Accept" recommendation.

Reviewer #1: All comments have been addressed

Reviewer #2: All comments have been addressed

2. Is the manuscript technically sound, and do the data support the conclusions?

Reviewer #1: Yes

Reviewer #2: Yes

3. Has the statistical analysis been performed appropriately and rigorously? 

Reviewer #1: Yes

Reviewer #2: Yes

4. Have the authors made all data underlying the findings in their manuscript fully available?

Reviewer #1: Yes

Reviewer #2: Yes

5. Is the manuscript presented in an intelligible fashion and written in standard English?

Reviewer #1: Yes

Reviewer #2: Yes

6. Review Comments to the Author

Reviewer #1: All the reviewer´s comments were addressed adequately and the manuscript was corrected accordingly. I have no further comments.

I recommend this manuscript to be accepted in the revised version presented herein.

Reviewer #2: As a reviewer, I sincerely appreciate the efforts made by you to thoughtfully address and resolve the comments. I am confident that the feedback provided in the initial review has contributed in improving both the quality of the data and the overall value of the information presented.

7. PLOS authors have the option to publish the peer review history of their article (what does this mean?). If published, this will include your full peer review and any attached files.

Reviewer #1: No

Reviewer #2: **Yes:**Umesh Chempanpathalil Gopalan

---

## [Editor Report · Acceptance letter]

PONE-D-25-68574R1

PLOS One

Dear Dr. Buddhirongawatr,

I'm pleased to inform you that your manuscript has been deemed suitable for publication in PLOS One. Congratulations! Your manuscript is now being handed over to our production team.

Kind regards,

on behalf of

Prof. Vincenzo Lionetti

Academic Editor

PLOS One